

# Implementing the Nitrogen cycle into the dynamic global vegetation, hydrology and crop growth model LPJmL (version 5)

Werner von Bloh[1], Sibyll Schaphoff[1], Christoph Müller[1], Susanne Rolinski[1], Katharina Waha[1,2], and Sönke Zaehle[3]

[1]Potsdam Institute for Climate Impact Research, P.O. Box 60 12 03, 14412 Potsdam, Germany
[2]CSIRO Agriculture & Food, 306 Carmody Rd, St. Lucia QLD 4067, Australia
[3]Max Planck Institute for Biogeochemistry, P.O. Box 60 01 64, 07701 Jena, Germany

*Correspondence to:* Werner von Bloh (bloh@pik-potsdam.de)

**Abstract.** The well-established dynamical global vegetation, hydrology, and crop growth model LPJmL is extended by a terrestrial nitrogen cycle to account for nutrient limitations. In particular, processes of soil nitrogen dynamics, plant uptake, nitrogen allocation, response of photosynthesis and maintenance respiration to varying nitrogen concentrations in plant organs, and agricultural nitrogen management are included into the model. All new model features are described in full detail and results of a global simulation of the historic past (1901-2009) are presented for evaluation of the model performance. We find that implementation of nitrogen limitation significantly improves the simulation of global patterns of crop productivity. Regional differences in crop productivity, which had to be calibrated via a scaling of the maximum leaf area index can now largely be reproduced by the model, except for regions where fertilizer inputs and climate conditions are not the yield limiting factors.

## 1 Introduction

Dynamics of the terrestrial biosphere and the associated terrestrial carbon cycle are of central importance for Earth System science. Climate-carbon cycle feedbacks have become integral parts of Earth System Models (ESMs) for climate change projections. However, the terrestrial carbon cycle dynamic are not only driven by climate and carbon dioxide ($CO_2$) fertilization (Schimel et al., 2015; Norby et al., 2005), but also by land-use change (Müller et al., 2006, 2016; Arneth et al., 2017; Le Quéré et al., 2016) and vegetation dynamics (Müller et al., 2016, and references therein). Nutrient limitations, especially from nitrogen, are also important constraints on vegetation growth and the terrestrial carbon cycle: Smith et al. (2016) suggested that Earth System Models contributing to the CMIP5 data archive overestimate the response of net primary productivity to elevated $CO_2$ because the models largely miss the constraints from nutrient limitation. Also Wieder et al. (2015) find that nitrogen limitation may substantially reduce projected increases in net primary productivity (NPP) under climate change and elevated atmospheric $CO_2$ concentrations ($[CO_2]$), possibly even converting the terrestrial biosphere into a net carbon source by the end of the 21st century. Over the last decade, nitrogen limitation has been increasingly accounted for in dynamic global vegetation (DGVM) and ESMs (Thornton et al., 2007; Gerber et al., 2010; Zaehle et al., 2010b; Smith et al., 2014). The Lund Potsdam Jena managed Land (LPJmL) dynamic global vegetation, hydrology and crop growth model has been widely applied to re-





search questions on the terrestrial carbon cycle, hydrology and agricultural production (Schaphoff et al., 2017b, and references therein) and performed similar to other dynamic vegetation models (Friend et al., 2014; Warszawski et al., 2013; Chang et al., 2017), hydrology models (Schewe et al., 2014) and crop models (Müller et al., 2017). However, LPJmL so far did not explicitly account for nutrient limitations. We here extend the LPJmL model to cover the terrestrial nitrogen cycle, by explicitly adding

processes of soil nitrogen dynamics, plant uptake, nitrogen allocation, response of photosynthesis and maintenance respiration to variable nitrogen concentrations in plant organs, and agricultural nitrogen management. We describe all new model features in full detail and present results of a global simulation of the historic past (1901-2009) that we use to evaluate model performance.

## 2 Model description

The Model description focuses on the nitrogen (N) dependent part of the model. A general description of the LPJmL model is supplied by Sitch et al. (2003); Bondeau et al. (2007); Schaphoff et al. (2017b, a). Note that Schaphoff et al. (2017b, a) provide the most comprehensive model description available, which includes a few model features that have been added to the model after the development of the N modules had begun and which are thus not part of the LPJmL 5 version described here. These include several minor amendments of the code as well as the updated grass allocation scheme (Rolinski et al., 2017) and the

updated phenology scheme for the natural vegetation (Forkel et al., 2014).

In the predecessor version LPJmL 3.5, all organic matter pools (vegetation, soil) were represented as carbon pools. We now also implemented a corresponding N pool for each of these carbon pools as well as pools for inorganic reactive N forms ($NH_4^+$, $NO_3^-$) in the soil (Fig. 1). In the following sections we describe the implementation of the plants' N demand, uptake, allocation, the effects of N limitation on photosynthesis and maintenance respiration as well as N inputs, transformations and

20 losses in/from soils. All processes are computed at a daily time step, except for fire events (annual) and the allocation of carbon and N in plants, which is computed daily only for crops but annually for natural vegetation and before each harvest event for managed grasslands.

### 2.1 Nitrogen demand

Daily photosynthesis and monthly maximum carboxylation capacity ($V_{max}$) are computed based on absorbed photosyntheti-

25 cally active radiation (apar) and canopy conductance reflecting the level of water stress (Sitch et al., 2003). This water-stressed carboxylation capacity $V_{max}$ determines the demand for N of trees, grasses and crops in the leaves. Depending on plant functional type (PFT)-specific requirements for $V_{max}$ the N demand of leaf, $N_{leaf}$ (gN m$^{-2}$), is calculated according to Smith et al. (2014) based on Haxeltine and Prentice (1996) as

$$N_{leaf} = 25 \cdot 0.02314815/\text{daylength} \cdot V_{max} \cdot \exp(-0.02 \cdot (T - 25)) \cdot f_{LAI}(LAI) + 0.00715 \cdot C_{leaf} \qquad (1)$$

where $C_{leaf}$ is the actual leaf carbon content (gC m$^{-2}$) and daylength is the duration of daylight (h). The function $f_{LAI}(LAI)$ is a modifier dependent on current leaf area index (LAI) accounting for a stronger leaf N content decline with canopy depth





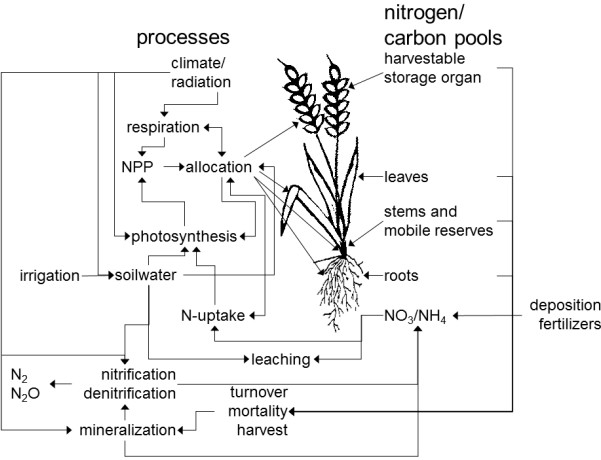

**Figure 1.** Carbon and nitrogen pools and associated processes for the example of crops.

compared to incoming sunlight:

$$f_{\mathrm{LAI}}(\mathrm{LAI}) = \begin{cases} \max(0.1, \mathrm{LAI}) & \text{for } \mathrm{LAI} < 1 \\ \exp(0.08 \cdot \min(\mathrm{LAI}, 7)) & \text{otherwise} \end{cases} \qquad (2)$$

The pre-factor $0.12$ in the exponential term of Smith et al. (2014) has been replaced by $0.08$ for two reasons. First, we find that canopy C:N ratios are too low for the original value. Second, the computed values for the average leaf C:N ratio of the canopy should monotonically increase with LAI, whereas they decline again at higher LAI. This unwanted decline is not completely prevented with our pre-factor of $0.08$ but much weaker and occurs only at much higher LAI values than in the original implementation (see SI Fig. S1). We choose a maximum of $\mathrm{LAI} = 7$ and for $\mathrm{LAI} < 1$ a linear decrease to avoid too high respiration rates at low LAI levels, where C:N ratios would become very small otherwise. Daily gross photosynthesis $A_{\mathrm{gd}}$ depends on light-limited photosynthesis rate $J_E$ and Rubisco-limited photosynthesis rate $J_C$:

$$A_{\mathrm{gd}} = \left( J_E + J_C - \sqrt{(J_E + J_C)^2 - 4 \cdot \theta \cdot J_E \cdot J_C} \right) / (2 \cdot \theta) \cdot \mathrm{daylength}, \qquad (3)$$

where $\theta$ is the shape parameter describing the co-limitation of light and Rubisco activity. The value of $\theta$ of LPJmL 3.5 has been changed from $0.7$ to $0.9$ which is in better agreement with Collatz et al. (1990) and results in lower Rubisco demand to reach the light-limited photosynthesis rate (see SI Fig. S2).

Because the allocation of carbon and nitrogen for grass and tree PFTs is done on a yearly time interval the actual carbon stored in leaves $C_{\mathrm{leaft},t}$ at time $t$ of the current year is calculated from the carbon stored in leaves at the end of the previous year $C_{\mathrm{leaf}}$:

$$C_{\mathrm{leaf},t} = C_{\mathrm{leaf}} + f_{\mathrm{leaf}} \cdot \sum_{t'=1}^{t} \mathrm{NPP}_{t'}, \qquad (4)$$





where $\sum_{t'} \text{NPP}_{t'}$ is the accumulated biomass increment and $f_{\text{leaf}}$ is the fraction of biomass that was allocated to leaves at the end of the previous year. Then the total N demand is determined by the actual ($t$) carboxylation-based demand for N in leaves $N_{\text{leaf},t}$ (see Eq. (1)), the current N content of the other organs (roots $N_{\text{root}}$ and sapwood $N_{\text{sapwood}}$ for trees) and the approximated N demand for the newly accumulated NPP (Eq. (5)). For this approximation, we use the allocation shares of the previous year ($f_{\text{root}}$, $f_{\text{sapwood}}$):

$$N_{\text{demand},t} = N_{\text{leaf},t} + N_{\text{root}} + N_{\text{sapwood}} + \frac{N_{\text{leaf}}}{C_{\text{leaf}}} \cdot (f_{\text{root}}/R_1 + f_{\text{sapwood}}/R_2) \cdot \sum_{t'=1}^{t} \text{NPP}_{t'}, \tag{5}$$

where $R_1$, $R_2$ are the prescribed PFT-specific C:N ratios of roots and sapwood relative to the leaf C:N ratio (Table 2). The daily allocation scheme of crops enables the calculation of nitrogen demand by using the carbon compartments itself. Plants maintain a store of labile N, $N_{\text{store}}$ (gN m$^{-2}$), to buffer fluctuations between N demand and supply from the soil mineral N pool (Smith et al., 2014). N demand is therefore increased by a factor of $k_{\text{store}} = 1.15$ for trees and of $k_{\text{store}} = 1.3$ for grass and crops. Thus, the optimum N uptake fulfilling the demand $N_{\text{uptake,opt}}$ can be calculated from the demand increment:

$$N_{\text{uptake,opt}} = (N_{\text{demand},t} - N_{\text{demand},t-1}) \cdot k_{\text{store}} \tag{6}$$

## 2.2 Nitrogen uptake

The mechanism for uptake of N ($N_{\text{uptake}}$ in gN m$^{-2}$ d$^{-1}$) is the same for trees, crops and grasses. Following Smith et al. (2014), plant $N_{\text{uptake}}$ is determined by soil mineral N concentrations, fine root mass, soil temperature and porosity, and plant demand for N. This is computed for all soil layers individually and summed up to compute overall N uptake:

$$N_{\text{uptake}} = \sum_{l=1}^{n_{\text{soillayer}}} 2 \cdot N_{\text{up,root}} \cdot f_N(N_{\text{avail},l}) \cdot f_T(T_{\text{soil},l}) \cdot f_{\text{NC}}(\text{NC}_{\text{plant}}) \cdot C_{\text{root}} \cdot \text{rootdist}_l, \tag{7}$$

where $N_{\text{up,root}}$ is the maximum N uptake rate per unit fine root mass in each layer, $f_N(N_{\text{avail}})$ parameterizes the dependence on available N, $f_T(T_{\text{soil}})$ parameterizes the temperature dependence, $f_{\text{NC}}$ parameterizes the dependence on plant N:C ratio, $C_{\text{root}}$ is the carbon stored in the roots, $n_{\text{soillayer}}$ is the number of soil layers ($n_{\text{soillayer}} = 6$) and $\text{rootdist}_l$ determines the fraction of roots in each layer. $N_{\text{up,root}}$ is $2.8 \times 10^{-3}$ gN gC$^{-1}$ d$^{-1}$ for trees and $5.51 \times 10^{-3}$ gN gC$^{-1}$ d$^{-1}$ for crops and grasses (Smith et al., 2014). The available N is the sum of $\text{NO}_3^-$ and $\text{NH}_4^+$ in the soil layer $l$:

$$N_{\text{avail},l} = \text{NO}_{3,\text{soil},l}^- + \text{NH}_{4,\text{soil},l}^+ \tag{8}$$

The function $f_N$ can be parameterized as a Michaelis-Menten kinetics:

$$f_N(N_{\text{avail},l}) = k_{N,\text{min}} + \frac{N_{\text{avail,l}}}{N_{\text{avail},l} + K_{N,\text{min}} \cdot \theta_{\text{max}} \cdot d_{\text{soil},l}}, \tag{9}$$

where $d_{\text{soil},l}$ is the soil column depth (m), $\theta_{\text{max}}$ is the soil type specific fractional pore space (dimensionless), $K_{N,\text{min}}$ is 1.48 gN m$^{-3}$ for woody and 1.19 for grassy PFTs (half saturation concentration of fine root N uptake), and $k_{N,\text{min}}$ (dimensionless)





is 0.05 , which is the basal rate of N uptake that is not associated with Michaelis-Menten kinetics. The function $f_{\mathrm{NC}}(\mathrm{NC}_{\mathrm{plant}})$ is from Zaehle et al. (2010b):

$$f_{\mathrm{NC}}(\mathrm{NC}_{\mathrm{plant}}) = \frac{\mathrm{NC}_{\mathrm{leaf,high}} - \mathrm{NC}_{\mathrm{plant}}}{\mathrm{NC}_{\mathrm{leaf,high}} - \mathrm{NC}_{\mathrm{leaf,low}}}, \tag{10}$$

where $\mathrm{NC}_{\mathrm{leaf,low}}$ and $\mathrm{NC}_{\mathrm{leaf,high}}$ are the lower and upper limits of N:C ratios and $\mathrm{NC}_{\mathrm{plant}}$ is the actual plant N:C ratio. The lower and upper limits $\mathrm{NC}_{\mathrm{leaf,low}}$ and $\mathrm{NC}_{\mathrm{leaf,high}}$ are derived from the TRY database (Kattge et al., 2011). Their reciprocal C:N values for each PFT are shown in Table 1. The actual plant N:C ratio is calculated according to

$$\mathrm{NC}_{\mathrm{plant}} = \frac{N_{\mathrm{leaf}} + N_{\mathrm{root}}}{C_{\mathrm{leaf}} + C_{\mathrm{root}}} \tag{11}$$

The temperature function $f_T$ for N uptake is given by Thornley (1991):

$$f_T(T_{\mathrm{soil},l}) = (T_{\mathrm{soil},l} - T_0) \cdot (2 \cdot T_m - T_0 - T_{\mathrm{soil},l})/(T_r - T_0)/(2 \cdot T_m - T_0 - T_r), \tag{12}$$

where $T_0 < T_r < 2 \cdot T_m - T_0$. For the chosen $T_m = 15°$C, $T_r = 15°$C and $T_0 = -25°$C, the maximum of 1 is reached at 15° and the function is positive above -25°C.

The root distribution $\mathrm{rootdist}_l$ is calculated from the PFT-specific root distribution $\mathrm{rootdist}_{PFT_l}$ and the thawing depth. If the soil depth of the layer $l$ is greater than the thawing depth then $\mathrm{rootdist}_l$ is reduced accordingly. The non-zero $\mathrm{rootdist}_l$ are rescaled so that their sum is normalized to one, accounting for the modified root distribution under freezing conditions. Soil $\mathrm{NH}_4^+$ and soil $\mathrm{NH}_3^-$ pools are reduced accordingly every simulation day $t$:

$$\mathrm{NO}_{3,\mathrm{soil},l,t+1}^- = \mathrm{NO}_{3,\mathrm{soil},l,t}^- \cdot \left(1 - \mathrm{rootdist}_l \cdot \frac{N_{\mathrm{uptake}}}{\sum_{l=1}^{n_{\mathrm{soillayer}}} N_{\mathrm{avail},l}}\right) \tag{13}$$

$$\mathrm{NH}_{4,\mathrm{soil}_l,t+1}^+ = \mathrm{NH}_{4,\mathrm{soil}_l,t}^+ \cdot \left(1 - \mathrm{rootdist}_l \cdot \frac{N_{\mathrm{uptake}}}{\sum_{l=1}^{n_{\mathrm{soillayer}}} N_{\mathrm{avail},l}}\right) \tag{14}$$

### 2.3 Determination of the N limitation scalar

For trees, grass and crops, the N limitation scalar $v_{\mathrm{scal}}$ is calculated by ratio of N demand $N_{\mathrm{uptake,opt}}$ and actual N uptake:

$$v_{\mathrm{scal}} = \min(N_{\mathrm{uptake}}/N_{\mathrm{uptake,opt}}, 1) \tag{15}$$

The scalar $v_{\mathrm{scal}}$ is used to account for N limitation in the allocation of N to different plant organs (section 2.5) and is computed as the growing season mean, which is re-initialized to zero every year for natural vegetation and at sowing for crops.

### 2.4 Net Primary Production under N limitation

To calculate the limitation by N availability, N stress is calculated after determining water stress on photosynthesis. If N demand cannot be fulfilled by N uptake, carboxylation capacity $V_{\mathrm{max}}$ has to be reduced. The reduced $V_{\mathrm{max}}$ is determined by solving Eq. (1) for $V_{\mathrm{max}}$. From this reduced $V_{\mathrm{max}}$, the actual photosynthesis rate can be calculated (Fig. 2). The gross primary production (GPP) derived from the actual photosynthesis rate is reduced by leaf, root and sapwood (for tree PFTs) respiration




**Table 1.** PFT-specific minimum and maximum leaf C:N ratios, based on the TRY data base (Kattge et al., 2011) with data from Kurokawa and Nakashizuka (2008); Garnier et al. (2007); Penuelas et al. (2010a); Fyllas et al. (2009); Loveys et al. (2003); Han et al. (2005); Ordonez et al. (2010); Atkin et al. (1999); White et al. (2000); Xu and Baldocchi (2003); Freschet et al. (2010a, b); Laughlin et al. (2010); Niinemets (2001, 1999); Willis et al. (2010); Baker et al. (2009); Patino et al. (2009); Pakeman et al. (2009, 2008); Fortunel et al. (2009); Penuelas et al. (2010b); Cornelissen et al. (1996, 1997, 2004); Quested et al. (2003); Sardans et al. (2008b, a); Ogaya and Penuelas (2003, 2006, 2007, 2008)

| Functional type | $CN_{leaf,low}$ | $CN_{leaf,high}$ | Source |
|---|---|---|---|
| Tropical broadleaved evergreen tree | 15.6 | 46.2 | TRY data |
| Tropical broadleaved raingreen tree | 15.4 | 34.6 | TRY data |
| Temperate needleleaved evergreen tree | 31.8 | 63.8 | TRY data |
| Temperate broadleaved evergreen tree | 15.6 | 46.2 | TRY data |
| Temperate broadleaved summergreen tree | 15.4 | 34.6 | TRY data |
| Boreal needleleaved evergreen tree | 31.8 | 63.8 | TRY data |
| Boreal broadleaved summergreen tree | 15.4 | 34.6 | TRY data |
| Boreal needleleaved summergreen tree | 18.4 | 36.9 | TRY data |
| C3 perennial grass | 10.5 | 37.9 | White et al. (2000) |
| C4 perennial grass | 17.4 | 66.9 | White et al. (2000) |
| Bioenergy tropical tree | 15.6 | 46.2 | TRY data |
| Bioenergy temperate tree | 15.4 | 34.6 | TRY data |
| Bioenergy C4 grass | 17.4 | 66.9 | White et al. (2000) |
| Crops | 14.3 | 58.8 | White et al. (2000) |

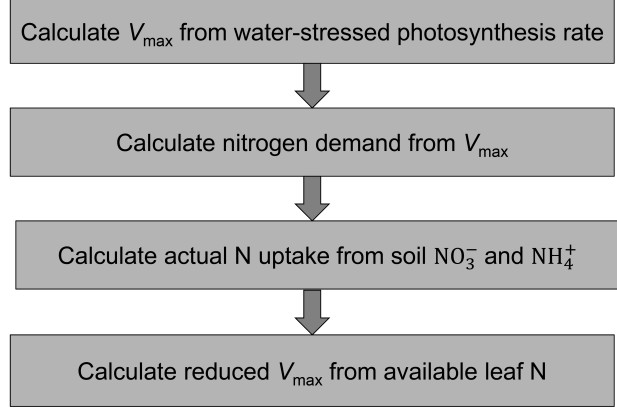

**Figure 2.** Calculation of N stress of plants.





$R_{\text{leaf}}$, $R_{\text{root}}$, and $R_{\text{sapwood}}$. Respiration rates of roots and sapwood is assumed to be linearly dependent on the N:C ratio of the corresponding pool, whereas the respiration rate of leaves ($R_{\text{leaf}}$) is a fraction (1.5% for C3 plants, 3.5% for C4 plants) of $V_{\text{max}}$ (Sitch et al., 2003):

$$R_{\text{root}} \quad = \quad k_{\text{resp}}(T_{\text{soil}}) \cdot N_{\text{root}} \tag{16}$$

$$R_{\text{sapwood}} \quad = \quad k_{\text{resp}}(T_{\text{air}}) \cdot N_{\text{sapwood}}, \tag{17}$$

where $k_{\text{resp}}(T)$ is a temperature dependent respiration rate (gC gN$^{-1}$ d$^{-1}$) (as in Sitch et al., 2003). Therefore higher N:C ratios lead to a reduction in net primary production (NPP), which is computed as:

$$\text{NPP} = \text{GPP} - R_{\text{growth}} - R_{\text{leaf}} - R_{\text{root}} - R_{\text{sapwood}}, \tag{18}$$

where $R_{\text{growth}}$ is 25% of GPP and $R_{\text{sapwood}}$ is zero for all non-woody plants.

## 2.5 Nitrogen allocation and turnover in plants

Carbon allocation to plant compartments follows functional and allometric rules as described by Sitch et al. (2003) and is computed annually for natural vegetation and daily for crops (Bondeau et al., 2007). The allocation rules account for the functional relationships that leaf area needs to be supported by sufficient sapwood (in trees) and fine root biomass. Fine root biomass increases relative to leaf biomass under water stress and also under nitrogen limitation. The allometric rules specify the relationship of stem diameter to plant height and crown diameter (Sitch et al., 2003). Plants require N in varying amounts to satisfy organ-specific C:N ratios. Leaf-N content is determined by the photosynthetic potential and structural requirements and can vary within PFT-specific limits of C:N-ratios. The PFT-specific range of possible C:N-ratios is based on the TRY data base (Kattge et al., 2011, Table 1).

The allocation of N ($N_{\text{inc}}$) to plant compartments follows the allocation rules for carbon and ensures the distribution between the plant compartments as established with the relative ratios given for the C:N ratio of, e.g., roots in comparison to leaves ($\text{CN}_{\text{root}}$ / $\text{CN}_{\text{leaf}}$). These relative ratios for natural vegetation are taken from Friend et al. (1997, Table 4).

For crops the C:N ratios for the storage organ are derived from Bodirsky et al. (2012). For this average crop functional type-specific leaf C:N ratios as simulated by LPJmL 5 were used to estimate the factors that relate leaf C:N ratios to storage organ C:N ratios (Table 2). The allocation scheme follows the algebraic solution of the following set of equations when there are $n$ plant compartments:

$$\frac{N_1 + a_1 \cdot N_{\text{inc}}}{C_1} \quad = \quad R_1 \cdot \frac{N_2 + a_2 \cdot N_{\text{inc}}}{C_2} \tag{19a}$$

$$\frac{N_1 + a_1 \cdot N_{\text{inc}}}{C_1} \quad = \quad R_2 \cdot \frac{N_3 + a_3 \cdot N_{\text{inc}}}{C_3} \tag{19b}$$

$$\vdots$$

$$\frac{N_1 + a_1 \cdot N_{\text{inc}}}{C_1} \quad = \quad R_{n-1} \cdot \frac{N_n + a_n \cdot N_{\text{inc}}}{C_n} \tag{19c}$$

$$\sum_{i=1}^{n} a_i \quad = \quad 1 \tag{19d}$$





**Table 2.** C:N ratios relative to the leaf C:N ratio $R_i$ for the different plant compartments.

| Plant | Root $R_1$ | Sapwood $R_2$ | Storage organ $R_3$ | Pool $R_4$ |
|---|---|---|---|---|
| Tree | 1.16 | 6.9 | | |
| Grass | 1.16 | | | |
| Temperate cereals | 1.16 | | 0.99 | 3 |
| Rice | 1.16 | | 1.30 | 3 |
| Maize | 1.16 | | 0.83 | 3 |
| Tropical cereals | 1.16 | | 0.79 | 3 |
| Pulses | 1.16 | | 0.45 | 3 |
| Potatoes | 1.16 | | 1.74 | 3 |
| Sugar beet | 1.16 | | 4.46 | 3 |
| Tropical roots | 1.16 | | 3.27 | 3 |
| Sunflower | 1.16 | | 1.04 | 3 |
| Soybeans | 1.16 | | 0.42 | 3 |
| Groudnut | 1.16 | | 0.68 | 3 |
| Rapeseed | 1.16 | | 0.76 | 3 |
| Sugar cane | 1.16 | | 4.57 | 3 |

where $C_1, C_2, \ldots, C_n, N_1, N_2, \ldots, N_n$ are the C and N pools of plant compartments $1, 2, \ldots, n$, and $R_1, R_2, \ldots R_{n-1}$ are the relative C:N ratios in comparison to leaves. The system is solved for $a_1, a_2, \ldots a_n$ so that the relative ratios $R_1, \ldots, R_{n-1}$ are ensured. Thus, the model has to solve the equation system for $n = 2$ pools for grass, for $n = 3$ pools for trees and for $n = 4$ pools for crops.

If the N:C ratio for a pool is below the PFT-specific minimum N:C ratio allowed then the excess carbon is put into the litter pools. To avoid overly large C fluxes from excess carbon to the litter pools in N-limited environments, we have introduced a sink-limitation for photosynthesis of trees. For this, the excess carbon from the sapwood pool is stored in an additional carbon pool $C_{\text{excess}}$. If this excess pool is filled and if there is a minimum $C_{\text{sapwood}}$ pool of at least 1 kg m$^{-2}$, photosynthesis is downregulated by a scaling factor $s$ in the following year (Eq. (20)). At the end of the year, the newly acquired carbon (NPP)

and the $C_{\text{excess}}$ are allocated to the plant organs, according to the usual allocation rules. If all carbon can be allocated within allowed compartment-specific C:N ratios, the $C_{\text{excess}}$ pool is empty afterwards and photosynthesis no longer downregulated.

$$s = (1 + K_M) \cdot \frac{f}{f + K_M}, f = \min\left(1, \frac{N_{\text{sapwood}}}{C_{\text{sapwood}} + C_{\text{excess}}} \cdot \frac{R_2}{\text{NC}_{\text{leaf,low}}}\right), \qquad (20)$$

where $K_M = 0.1$ is the Michaelis constant of the Michaelis-Menten kinetics and $R_2$ is the relative C:N ratio of sapwood in respect to leaves.

Similar to water stress, we assume that plants allocate more biomass to roots under N limitation. For this, the leaf to root mass ratio lmtorm is modified by the minimum of the N limitation factor $v_{\text{scal}}$ and the water limitation factor $w_{\text{scal}}$. Both factors are





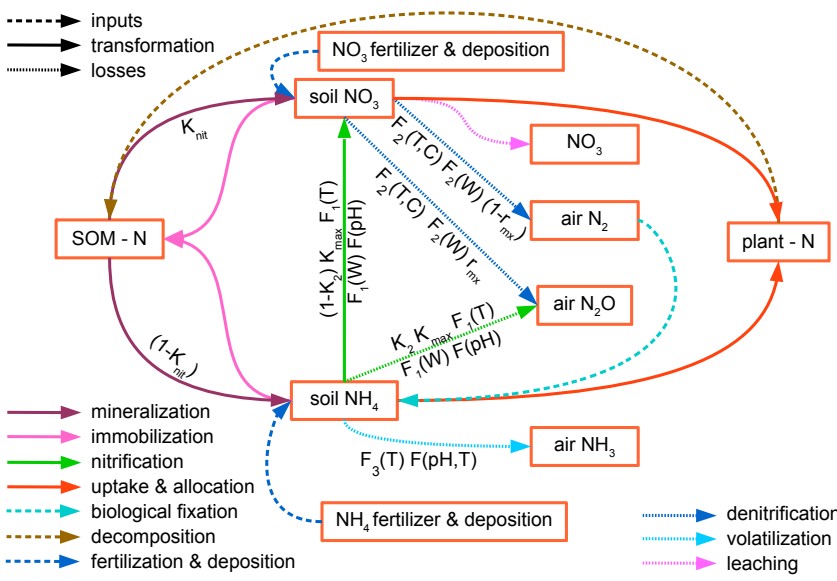

**Figure 3.** Nitrogen transformations and losses in soils. Pools and fluxes are denoted by boxes and arrows, respectively.

computed as growing season means with daily updates, i.e. for the entire calendar year for natural vegetation, between harvest events for managed grasslands, and since sowing for crops.

LPJmL employs PFT-specific turnover rates for living leaves and fine roots. At turnover the corresponding amount of carbon is moved into the litter pools, whereas not all of the associated N is disposed but remains in the plant. We assume that grasses and deciduous trees recover $k_{\mathrm{turn}} = 70\%$ of their N upon biomass turnover, whereas evergreen trees only recover $k_{\mathrm{turn}} = 20\%$. At turnover sapwood carbon is transformed into heartwood carbon. Not all nitrogen of the sapwood turnover is going into heartwood, only a fraction $f_{\mathrm{heartwood}} = 0.7$ of nitrogen is transformed.

## 2.6 Nitrogen transformation in soils

Nitrogen occurs in soils in different reactive forms, mainly organic forms, nitrate ($NO_3^-$) and ammonium ($NH_4^+$), which are represented by different pools in LPJmL5. Transformations between different forms of N in the soil are represented by mineralization, immobilization, nitrification, and denitrification. Losses from the soil are represented by the implemented nitrification, denitrification, leaching, and volatilization processes. The corresponding pools and fluxes are depicted in Fig. 3 and described, including their parameterization (see SI Table S1), in this section.



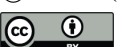

### 2.6.1 Mineralization of nitrogen

Mineralization of N from soil organic matter and decomposition of litter pools follows that of carbon as described by Schaphoff et al. (2013). First, for each soil layer the fluxes of carbon from the soil into the atmosphere are calculated and the respective fluxes of N, reflecting actual C:N ratios of the material, are transferred to the $NH_4^+$ (80%) and $NO_3^-$ (20%) soil pools of the corresponding soil layer.

Fluxes ($F$) of carbon and nitrogen for slow ($s$) and fast ($f$) pools ($P$) depend on parameters $k_{\text{soil10}}^f = 0.03$ and $k_{\text{soil10}}^s = 0.001$ (per year) and $R(T, M)$ as a function of temperature ($T$) and soil moisture ($M$) per soil layer ($l$).

$$F_l^x = \max(0, P_l^x \cdot (1 - \exp(-k_{\text{soil10}}^x \cdot R(T_l, M_l))), \ x \ \in \ (s, f), \tag{21}$$

where

$$R(T_l, M_l) = T_l \cdot (0.04021601 - 5.00505434 \cdot M_l^3 + 4.26937932 \cdot M_l^2 + 0.71890122 \cdot M_l) \tag{22}$$

The mineralization of soil $N_{\text{miner,soil},l}$ in soil layer $l$ is given by

$$N_{\text{miner,soil},l} = F_l^f + F_l^s \tag{23}$$

Whereas the mineralization fluxes of carbon go completely to the atmosphere as $CO_2$, mineralized N goes to the mineral pools, where they are subject to further transformation (Parton et al., 2001).

Decomposition of N in soil organic material ($N_{\text{decom}}$) consists of a mineralization part ($A_f = 0.7$, dimensionless) that forms $NH_4^+$ (80%) and $NO_3^-$ (20%), and a humification part ($1 - A_f$), in which organic N from the litter pool is transferred to the soil pools. The humification flux is divided into fluxes to slow ($s$) and fast ($f$) N soil pools ($P$); the parameter $F_f = 0.98$ (dimensionless) specifies the portion that goes to the fast soil pool.

$$P_{l,t+1}^f \quad = \quad P_{l,t}^f + F_f \cdot (1 - A_f) \cdot N_{\text{decom}} \cdot N_{\text{shift},l}^f \tag{24}$$
20
$$P_{l,t+1}^s \quad = \quad P_{l,t}^s + (1 - F_f) \cdot (1 - A_f) \cdot N_{\text{decom}} \cdot N_{\text{shift},l}^s, \tag{25}$$

where the annual shift rates $N_{\text{shift},l}^{s,f}$ describe the organic matter input from the different PFTs into the respective layer due to cryoturbation and bioturbation (Schaphoff et al., 2013).

Net mineralized material $N_{\text{miner,litter},l}$ is

$$N_{\text{miner,litter},l} = A_f \cdot N_{\text{decom}} \cdot (F_f \cdot N_{\text{shift},l}^f + (1 - F_f) \cdot N_{\text{shift},l}^s), \tag{26}$$

which adds for N to an intermediate N mineralization pool as well

$$N_{\text{miner},l} = N_{\text{miner,soil}, l} + N_{\text{miner,litter}, l} \tag{27}$$

A fraction of $K_{\text{nit}} = 0.2$ (dimensionless) of this pool is nitrified directly per day to $NO_3^-$ (see Eq. 2 in Parton et al. (2001)) and the fluxes are added to the soil pools of $NH_4^+$ and $NO_3^-$ (g m$^{-2}$).

$$NH_{4,\text{soil},l,t+1}^+ \quad = \quad NH_{4,\text{soil},l,t}^+ + (1 - K_{\text{nit}}) \cdot N_{\text{miner},l} \tag{28}$$
30
$$NO_{3,\text{soil},l,t+1}^- \quad = \quad NO_{3,\text{soil},l,t}^- + K_{\text{nit}} \cdot N_{\text{miner},l} \tag{29}$$



### 2.6.2 Nitrogen immobilization

Immobilization, i.e. the transformation of mineral N to organic N in soils, is determined per soil layer directly after soil and litter mineralization, following the CLM approach described by Gerber et al. (2010). If available mineral soil N is constraining immobilization, mineral N is first immobilized into the fast soil pool and then into the slow soil pool. The immobilized N $N_{\text{immo},l}$ is calculated according to

$$N_{\text{immo},l} = F_f \cdot (1 - A_f) \cdot (C_{\text{decom}}/\text{CN}_{\text{soil}} - N_{\text{decom}}) \cdot N_{\text{shift},l}^f \cdot \frac{N_{\text{sum},l}/d_{\text{soil},l}}{k_N + N_{\text{sum},l}/d_{\text{soil},l}}, \tag{30}$$

where $\text{CN}_{\text{soil}}$ is the desired soil C:N ratio of 15 (dimensionless) for all soil types, $d_{\text{soil},l}$ is the soil depth of layer $l$ in m, $k_N = 5 \times 10^{-3}$ (gN m$^{-3}$) is the half saturation concentration for immobilization in soils (Gerber et al., 2010), and $N_{\text{shift},l}^f$ is the parameter that determines the distribution of the humified organic matter in the topsoil to the different soil layers $l$ (Schaphoff et al., 2013). The available mineral N in the soil layer $l$ ($N_{\text{sum},l}$ in gN m$^{-2}$) is the sum of $\text{NH}_4^+$ and $\text{NO}_3^-$:

$$N_{\text{sum},l} = \text{NH}_{4,\text{soil},l}^+ + \text{NO}_{3,\text{soil},l}^- \tag{31}$$

The immobilized N ($N_{\text{immo},l}$) is added to the fast soil N pool of layer $l$ and subtracted from the $\text{NH}_4^+$ and $\text{NO}_3^-$ pools:

$$
\begin{aligned}
P_{\text{soil},l,t+1}^f &= P_{\text{soil},l,t}^f + \min(N_{\text{immo},l}, N_{\text{sum},l}) & (32)\\
\text{NH}_{4,\text{soil},l,t+1}^+ &= \text{NH}_{4,\text{soil},l,t}^+ - \text{NH}_{4,\text{soil},l,t}^+ \cdot \min(N_{\text{immo},l}/N_{\text{sum},l}, 1) & (33)\\
\text{NO}_{3,\text{soil},l,t+1}^- &= \text{NO}_{3,\text{soil},l,t}^- - \text{NO}_{3,\text{soil},l,t}^- \cdot \min(N_{\text{immo},l}/N_{\text{sum},l}, 1) & (34)
\end{aligned}
$$

The immobilization into the slow soil N pool ($P_{\text{soil},l,t+1}^s$) is computed accordingly as in Eq. (30) but with $(1 - F_f)$ instead of $F_f$.

### 2.6.3 Nitrification

Nitrogen fluxes from nitrification in the soil are modeled modified after Parton et al. (2001) with the schematic representation of a series of pipes for the main flow from $\text{NH}_4^+$ over $\text{NO}_3^-$ to $\text{N}_2$ from which $\text{N}_2\text{O}$ leaks in between. As suggested by Parton et al. (2001, equation 2), nitrification is computed as a fixed fraction of the mineralization flux (see 2.6.1) as well as an explicit transformation flux $F_{\text{NO}_3^-}$ from ammonium to nitrate in gN m$^{-2}$ d$^{-1}$, which is described here.

$$F_{\text{NO}_3^-} = K_{\max} \cdot F_1(T_{\text{soil}}) \cdot F_1(W_{\text{sat}}) \cdot F(\text{pH}) \cdot \text{NH}_{4,\text{soil}}^+, \tag{35}$$

where $\text{NH}_{4,\text{soil}}^+$ is the model-derived soil ammonium concentration (gN m$^{-2}$), $K_{\max}$ is the maximum nitrification rate of $\text{NH}_4^+$ ($K_{\max} = 0.1$ d$^{-1}$), $F_1(T_{\text{soil}})$ is the limiting function for temperature and $F_1(W_{\text{sat}})$ the corresponding limiting function for water saturation $W_{\text{sat}}$. Parton et al. (2001) show nitrification rates after data of Malhi and McGill (1982) in Table 3 without a formula. Using these data from three different sites in the US, Canada and Australia, we fitted a bell shaped function for the temperature dependence:

$$F_1(T_{\text{soil}}) = \exp(-(T_{\text{soil}} - a)^2/(2 \cdot b^2)), \tag{36}$$



where $a = 18.79°C$ and $b = 5.26$ give the best fist to the data (see SI Fig. S4). The function is applicable also for negative values.

The soil water response function $F_1(W_{\mathrm{sat}})$ is parameterized according to Doran et al. (1988) as described in Parton et al. (1996):

$$F_1(W_{\mathrm{sat}}) = \left(\frac{W_{\mathrm{sat}} - b_{\mathrm{nit}}}{a_{\mathrm{nit}} - b_{\mathrm{nit}}}\right)^{d_{\mathrm{nit}} \cdot (b_{\mathrm{nit}} - a_{\mathrm{nit}})/(a_{\mathrm{nit}} - c_{\mathrm{nit}})} \cdot \left(\frac{W_{\mathrm{sat}} - c_{\mathrm{nit}}}{a_{\mathrm{nit}} - c_{\mathrm{nit}}}\right)^{d_{\mathrm{nit}}}, \tag{37}$$

where $W_{\mathrm{sat}}$ is the water filled pore space of the soil, parameters $a_{\mathrm{nit}}$ to $d_{\mathrm{nit}}$ are given for sandy and medium soil (SI Table S1).

This soil pH function is based on Parton et al. (1996):

$$F(\mathrm{pH}) = 0.56 + \arctan(\pi \cdot 0.45 \cdot (-5 + \mathrm{pH}))/\pi \tag{38}$$

Soil pH values are taken from the WISE dataset (Batjes, 2000). Part of the N during the nitrification is lost to the atmosphere as nitrous oxide $N_2O$. Parton et al. (2001) assume that the $N_2O$ flux $F_{N_2O}$ (in $\mathrm{gN\ m^{-2}\ d^{-1}}$) is proportional to the nitrification rate with

$$F_{N_2O} = K_2 \cdot F_{NO_3^-}, \tag{39}$$

where $K_2$ is fraction of nitrified N lost as $N_2O$ flux ($K_2 = 0.02$). Finally, soil $NO_3^-$ and $NH_4^+$ are updated accordingly:

$$NO_{3,\mathrm{soil},l,t+1}^- = NO_{3,\mathrm{soil},l,t}^- + (1 - K_2) \cdot F_{NO_3^-} \tag{40}$$
$$NH_{4,\mathrm{soil},l,t+1}^+ = NH_{4,\mathrm{soil},l,t}^+ - F_{NO_3^-} \tag{41}$$

### 2.6.4 Denitrification

The reduction of $NO_3^-$ to $NO_2$ and $N_2$ is determined for each soil layer using the implementation in SWIM (Krysanova and Wechsung, 2000).

$$D_{NO_3^-} = F_2(W_{\mathrm{sat}}) \cdot F_2(T_{\mathrm{soil}}, C_{\mathrm{org}}) \cdot NO_{3,\mathrm{soil}}^-, \tag{42}$$

where $F_2(W_{\mathrm{sat}})$ is the water response function and $F_2(T, C)$ the soil temperature and carbon reaction function. The water response function depends on the water filled pore space $W_{\mathrm{sat}}$ in the following way:

$$F_2(W_{\mathrm{sat}}) = 6.664096 \times 10^{-10} \cdot \exp(21.12912 \cdot W_{\mathrm{sat}}) \tag{43}$$

The water response function shows a qualitatively similar behavior to Eq. 151 from SWIM while ensuring continuity (see SI Fig. S4). Parameters are fitted and adjusted so that for full soil water saturation, the value is not greater than 1. The soil temperature and carbon reaction function is parameterized according to:

$$F_2(T_{\mathrm{soil}}, C_{\mathrm{org}}) = 1 - \exp(-\mathrm{CDN} \cdot F_2(T_{\mathrm{soil}}) \cdot C_{\mathrm{org}}), \tag{44}$$

where $\mathrm{CDN} = 1.4$ is the shape coefficient (Arnold et al., 2012), $C_{\mathrm{org}}$ is the sum of the fast and slow C pools and $F_2(T_{\mathrm{soil}})$ is the soil temperature reaction function. $F_2(T_{\mathrm{soil}})$ is replaced by Equation C5 from Smith et al. (2014) which is only valid for





positive $T_{\text{soil}}$. The original function from SWAT approaches 1 for high temperatures whereas the function from Smith declines which seems more sensible. Equation C5 of Smith et al. (2014) is taken from Comins and McMurtrie (1993).

$$F_2(T_{\text{soil}}) = \begin{cases} 0.0326 & \text{for } T_{\text{soil}} \leq 0°\text{C} \\ 0.0326 + 0.00351 \cdot T_{\text{soil}}^{1.652} - (T_{\text{soil}}/41.748)^{7.19} & \text{for } 0°\text{C} < T_{\text{soil}} < 45.9°\text{C} \\ 0 & \text{for } T_{\text{soil}} \geq 45.9°\text{C} \end{cases} \tag{45}$$

Bessou et al. (2010) assume that the $N_2O$ flux from $NO_3^-$, $F_{N_2O}$ (in gN m$^{-2}$ d$^{-1}$), is proportional to the denitrification rate $D_{NO_3^-}$ with

$$F_{N_2O} = r_{\text{mx}} \cdot D_{NO_3^-}, \tag{46}$$

where $r_{\text{mx}} = 0.11$ is the fraction of denitrified N lost as $N_2O$ flux. The $N_2$ flux $F_{N_2}$ is then derived by

$$F_{N_2} = (1 - r_{\text{mx}}) \cdot D_{NO_3^-}, \tag{47}$$

The soil $NO_3^-$ pools have to be reduced by the denitrification flux:

$$NO_{3,\text{soil},l,t+1}^- = NO_{3,\text{soil},l,t}^- - D_{NO_3^-} \tag{48}$$

### 2.6.5 Nitrogen leaching and movement

Nitrate movement with water fluxes is simulated as in SWAT (Neitsch et al., 2002, 2005). Nitrate is assumed to be fully dissolved in water and moves with surface runoff, lateral runoff and percolation water. To compute the amount of nitrate transported with the water from a soil layer, we first calculate the concentration of nitrate in the mobile water. This concentration is then multiplied by the volume of surface runoff, lateral runoff or percolation water between soil layers or into the aquifer, respectively. The amount of nitrate leached depends on the climatic and soil conditions and on the type and intensity of soil management (e.g. plant cover, soil treatment, fertilization).

The concentration of nitrate in the mobile water $\text{conc}_{NO_3^-,\text{mobile},l}$ in layer $l$ (kgN m$^{-3}$) is:

$$\text{conc}_{NO_3^-,\text{mobile},l} = \frac{NO_{3,\text{soil},l}^- \cdot \left(1 - \exp\left(\frac{-w_{\text{mobile},l}}{(1-\theta)\cdot\text{SAT}_l}\right)\right)}{w_{\text{mobile},l}}, \tag{49}$$

where $NO_{3,\text{soil},l}^-$ is the content of nitrate in layer $l$ (gN m$^{-2}$), $w_{\text{mobile}}$ is the amount of mobile water in the layer (mm), $\theta = 0.4$ is the fraction of porosity from which anions are excluded (0.5 in Neitsch et al., 2002), and $\text{SAT}_l$ is the saturated water content of the soil layer (mm).

The mobile water $w_{\text{mobile},l}$ in the layer $l$ is the amount of water lost by surface runoff, lateral flow and percolation:

$$w_{\text{mobile},l} = \begin{cases} Q_{\text{surf}} + Q_{\text{lat},l=1} + w_{\text{perc},l=1} & \text{for } l = 1 \\ Q_{\text{lat},l} + w_{\text{perc},l} & \text{for } l > 1 \end{cases}, \tag{50}$$



where $Q_{\mathrm{surf}}$ is the surface runoff (only in top soil layer, mm), $Q_{\mathrm{lat},l}$ is the water discharged from the layer by lateral flow (mm) and $w_{\mathrm{perc},l}$ is the amount of water percolating to the underlying soil layer on a given day.

Finally, the amount of nitrate that is removed with surface runoff $\mathrm{NO_3^-}_{\mathrm{surf}}$ and lateral flow $\mathrm{NO_3^-}_{\mathrm{lat},l}$ is calculated as:

$$\mathrm{NO_3^-}_{\mathrm{surf}} = \beta_{\mathrm{NO_3^-}} \cdot \mathrm{conc}_{\mathrm{NO_3^-},\mathrm{mobile}} \cdot Q_{\mathrm{surf}} \tag{51}$$

$$\mathrm{NO_3^-}_{\mathrm{lat},l=1} = \beta_{\mathrm{NO_3^-}} \cdot \mathrm{conc}_{\mathrm{NO_3^-},\mathrm{mobile}} \cdot Q_{\mathrm{lat},l=1} \tag{52}$$

for the top layer and

$$\mathrm{NO_3^-}_{\mathrm{lat},l} = \mathrm{conc}_{\mathrm{NO_3^-},\mathrm{mobile},l} \cdot Q_{\mathrm{lat},l} \tag{53}$$

for the lower soil layers, where $\beta_{\mathrm{NO_3^-}}$ is the nitrate percolation coefficient. It controls the amount of $\mathrm{NO_3^-}$ removed from the surface layer in runoff relative to the amount removed via percolation (Neitsch et al., 2002). The value for $\beta_{\mathrm{NO_3^-}}$ can range from 0.01 to 1.0. For $\beta_{\mathrm{NO_3^-}} \to 0$, the concentration of nitrate in the runoff approaches 0. For $\beta_{\mathrm{NO_3^-}} = 1.0$, surface runoff has the same concentration of nitrate as the percolating water. We choose for $\beta_{\mathrm{NO_3^-}}$ a value of 0.4.

Nitrate moved to the lower soil layer with percolation $\mathrm{NO_3^-}_{\mathrm{perc},l}$ is calculated as:

$$\mathrm{NO_3^-}_{\mathrm{perc},l} = \mathrm{conc}_{\mathrm{NO_3^-},\mathrm{mobile}} \cdot w_{\mathrm{perc},l} \tag{54}$$

$\mathrm{NO_3^-}_{\mathrm{perc},l}$ is subtracted from current $\mathrm{NO_3^-}$ in the soil layer and added to the $\mathrm{NO_3^-}$ pool of the following soil layer:

$$\mathrm{NO_{3,soil},l,t+1}^- = \begin{cases} \mathrm{NO_{3,soil},l,t}^- - \mathrm{NO_3^-}_{\mathrm{perc},l} - \mathrm{NO_3^-}_{\mathrm{surf}} - \mathrm{NO_3^-}_{\mathrm{lat},l} & \text{for } l = 1 \\ \mathrm{NO_{3,soil},l,t}^- + \mathrm{NO_3^-}_{\mathrm{perc},l-1} - \mathrm{NO_3^-}_{\mathrm{perc},l} - \mathrm{NO_3^-}_{\mathrm{lat},l} & \text{for } l > 1 \end{cases} \tag{55}$$

### 2.6.6 Nitrogen volatilization

Volatilization of $\mathrm{NH_4^+}$ is parameterized according to Montes et al. (2009). A convective mass transfer model is applied where the flux varies with air temperature, air velocity over the surface, and the $\mathrm{NH_3}$ concentration gradient between the ammonium ($\mathrm{NH_4^+}$) in solution and in the air:

$$J_{\mathrm{NH_3}} = h_m \cdot \left([\mathrm{NH_3}]_{\mathrm{gas}} - [\mathrm{NH_3}]_{\mathrm{air}}\right), \tag{56}$$

where $J_{\mathrm{NH_3}}$ is the $\mathrm{NH_3}$ volatilization flux ($\mathrm{gNH_3}$-N m$^{-2}$ s$^{-1}$), $h_m$ is the convective mass transfer coefficient (m s$^{-1}$), $[\mathrm{NH_3}]_{\mathrm{gas}}$ is the concentration of gaseous $\mathrm{NH_3}$ in equilibrium with dissolved $\mathrm{NH_3}$ in solution ($\mathrm{gNH_3}$-N m$^{-3}$ air), and $[\mathrm{NH_3}]_{\mathrm{air}}$ is the concentration of $\mathrm{NH_3}$ in ambient air ($\mathrm{gNH_3}$-N m$^{-3}$ air), which is usually very small and can be neglected. The convective mass transfer coefficient $h_m$ is a function of temperature $T$ (in K), air velocity $U$ (in m s$^{-1}$) and characteristic length $L$ (in m) of the emitting surface:

$$h_m = 0.000612 \cdot U^{0.8} \cdot T^{0.382} \cdot L^{-0.2} \tag{57}$$

The concentration of gaseous $\mathrm{NH_3}$ in equilibrium with the dissolved $\mathrm{NH_3}$ is determined using Henry's law. The Henry's law $K_h$ constant relates the concentration of dissolved $\mathrm{NH_3}$ in water to an equilibrium concentration of $\mathrm{NH_3}$ in the air:

$$K_h = \frac{[\mathrm{NH_3}]_{\mathrm{gas}}}{[\mathrm{NH_3}]_{\mathrm{solution}}} \tag{58}$$



The Henry's law constant $K_h$ can be parameterized as a function of air temperature $T_{\mathrm{air}}$ (in K):

$$K_h = K_h(T_{\mathrm{air}}) = (0.2138/T_{\mathrm{air}}) \cdot 10^{6.123 - 1825/T_{\mathrm{air}}} \tag{59}$$

The fraction of total ammonical N present as $NH_3$ can be estimated using equilibrium thermodynamic principles:

$$f_{NH_3} = \frac{[NH_3]_{\mathrm{solution}}}{[NH_3]_{\mathrm{solution}} + [NH_4^+]_{\mathrm{solution}}} \tag{60}$$

$$= \frac{1}{1 + \frac{[H^+]}{K_a}} = \frac{1}{1 + \frac{10^{-pH}}{K_a}}, \tag{61}$$

where $K_a$ is the dissociation constant, $[H^+]$ is the proton concentration in solution, and $pH = -\log([H^+])$. The dissociation constant $K_a$ is parameterized as a function of temperature $T$ (in K):

$$K_a = K_a(T) = 10^{0.05 - 2788/T} \tag{62}$$

Then the volatilization flux $F_{\mathrm{vol}}$ (in gN m$^{-2}$ d$^{-1}$) is calculated according to

$$F_{\mathrm{vol}} = 86400 \cdot h_m(U, T, L) \cdot K_h(T) \cdot \frac{1}{1 + \frac{10^{-pH}}{K_a(T)}} \cdot NH_{4,\mathrm{soil},l=1}^+ / d_{\mathrm{soil},l=1} \tag{63}$$

and soil $NH_4^+$ is reduced in the top layer $l = 1$ accordingly:

$$NH_{4,\mathrm{soil},l=1,t+1}^+ = NH_{4,\mathrm{soil},l=1,t}^+ - F_{\mathrm{vol}} \tag{64}$$

### 2.7 Nitrogen and fire

Fire creates emissions of $N_2O$ and $NO_x$ and leaves nutrient rich ashes as well as charcoal. Following Gerber et al. (2010), the flux of N due to fire is divided between atmospheric emission and ash introduction to the nitrate pool of the upper soil layer $NO_{3,\mathrm{soil},l=1}^-$.

$$N_{\mathrm{fire}} = C_{\mathrm{fire}} \cdot N_{\mathrm{pool}} / C_{\mathrm{pool}} \tag{65}$$

$$N_{\mathrm{emission}} = (1 - q_{\mathrm{ash}}) \cdot N_{\mathrm{fire}} \tag{66}$$

$$NO_{3,\mathrm{soil},l=1,t+1}^- = NO_{3,\mathrm{soil},l=1,t}^- + q_{\mathrm{ash}} \cdot N_{\mathrm{fire}}, \tag{67}$$

where $q_{\mathrm{ash}} = 0.45$ is the fraction of N going into the top soil layer $NO_3^-$.

### 2.8 Biological N-fixation

The biological fixation of N occurs at all stands with an exception for agricultural stands. There, is it applied only for the nodulating leguminous crops pulses and soybean. For these two crops, biological N-fixation (BNF) is simply the difference between N demand and N uptake, basically first using the easily plant-available N from the soils and then fixing extra N at no extra cost. For natural vegetation and grasslands, the function from Cleveland et al. (1999) is applied depending on the 20-year



average annual evapotranspiration $\mathrm{etp}$ (in mm yr$^{-1}$). BNF (in gN m$^{-2}$ d$^{-1}$) is assumed to only occur if there is a minimum root biomass of 20 gC m$^{-2}$. All N fixed by BNF is assumed to enter the system as ammonium in the upper soil layer ($l = 1$).

$$\mathrm{BNF} = \begin{cases} \max(0, (0.0234 \cdot \mathrm{etp} - 0.172)/10/365) & \text{if } C_{\mathrm{root}} > 20 \text{ gC m}^{-2} \\ 0 & \text{otherwise} \end{cases} \tag{68}$$

$$\mathrm{NH}^{+}_{4,\mathrm{soil},l=1,t+1} = \mathrm{NH}^{+}_{4,\mathrm{soil},l=1,t} + \mathrm{BNF} \tag{69}$$

The function gives linearly increasing values which are positive for $\mathrm{etp} > 7.35$ and are set to zero otherwise. Note that in Zaehle et al. (2010a) this function is also cited in the supplementary material but with a positive intercept which is not the original equation from Cleveland et al. (1999).

## 2.9 Nitrogen fertilization of crops

Fertilizer is applied at sowing and when the amount of fertilizer is larger than 5 gN m$^{-2}$), only half of the fertilizer is applied at sowing. The second application occurs when the phenological stage of the crop development $\mathrm{fphu}$ exceeds $0.4$ to avoid large loss fluxes (leaching, volatilization, nitrification, denitrification) when fertilizing large amounts of N at the beginning of the season.

Nitrogen fertilizer is assumed to be ammonium nitrate ($\mathrm{NH}_4\mathrm{NO}_3$), so half of the applied rate is put into the top soil layer nitrate pool ($\mathrm{NO}^{-}_{3,\mathrm{soil},l=1}$) and the other half into the top soil layer ammonium pool ($\mathrm{NH}^{+}_{4,\mathrm{soil},l=1}$).

## 3 Model setup

For the assessment of model performance, we focus on the historic period 1901-2009. We conduct 6 different sets of simulations, two simulations with the carbon-only predecessor model version LPJmL 3.5 and four with the newly implemented nitrogen version LPJmL 5. Both model versions are used for a standard historic simulation, with dynamic land-use change, referred to as *LPJmL35* and *LPJmL5* respectively, as well as for a simulation without human land use, where potential natural vegetation (PNV) is simulated on the entire land surface. These runs are referred to as *LPJmL35-PNV* and *LPJmL5-PNV*. For analyzing the current N-limitation, we also conduct a simulation with dynamic land use, but with unlimited N supply (*LPJmL5-nL*) and one with no fertilizer application (*LPJmL5-nF*).

### 3.1 Model input

Model simulations are driven with observational monthly input data on monthly precipitation from the Global Precipitation Climatology Centre (GPCC Full Data Reanalysis Version 7.0, Becker et al., 2013) and daily mean temperatures from the Climatic Research Unit (CRU TS version 3.23, University of East Anglia Climatic Research Unit; Harris, 2015; Harris et al., 2014). Radiation data, shortwave downward and net downward longwave, are provided by reanalysis data from ERA-Interim (Dee et al., 2011). Monthly precipitation is allocated to individual days of the corresponding month by deriving the number of wet days per month synthetically as suggested by New et al. (2000).





Land-use input is derived from MIRCA2000 (Portmann et al., 2010) using the maximum monthly growing areas per crop and grid cell combined with the extent of areas equipped for irrigation (Siebert et al., 2015). HYDE3 (Klein Goldewijk and van Drecht, 2006) gives the relative changes of cropland and pasture extent backward to 1700. Further information are given by Fader et al. (2010).

The global dataset "Simulated Topological Network" (STN-30) drainage direction map (Vorosmarty and Fekete, 2011) gives transport directions of the river routing scheme. We use the GRanD database (Lehner et al., 2011), which provides detailed information on water reservoirs that includes information on storage capacity, total area and main purpose. Furthermore, information on natural lakes are obtained from Lehner and Döll (2004).

Nitrogen deposition is based on the ACCMIP database (Lamarque et al., 2013) for $NO_3^-$ and $NH_4^+$ separately, which is
applied daily to the corresponding mineral N pools of the top soil layer. Dry and wet deposition is not distinguished. Soil pH data are taken from the WISE dataset (Batjes, 2000). Fertilizer data is crop-specific, but static in time. We use the data supplied by the Global Gridded Crop Model Intercomparison (GGCMI phase 1 Elliott et al., 2015), which is based on gridded mineral fertilizer data (Mueller et al., 2012) and manure data (Potter et al., 2010) from which 60% is assumed to be plant available (Elliott et al., 2015).

**3.2   Model initialization, spin up and equilibration of soil**

All carbon and water pools are initialized to zero except soil water, soil carbon and soil temperatures, which are computed from a 30 yr averaged climate. Then a spin-up simulation of 5000 years is performed to bring permafrost extent, vegetation patterns and carbon stocks into dynamic equilibriums. The long spin-up time is necessary for reaching these equilibrium states in the permafrost regions (Schaphoff et al., 2013).

Soil N pools (organic and mineral) are initialized with assumptions to allow for initial vegetation growth. Organic N pools (slow, fast) as well as mineral N pools ($NO_3^-$ and $NH_4^+$) are set to $10^4$ gN m$^{-2}$. After 1320 simulation years, vegetation composition is assumed to have reached an equilibrium (Schaphoff et al., 2013) and litterfall is tracked for another 3680 years to allow for estimating soil carbon and soil N stocks. Based on these estimates for carbon and N fluxes under equilibrium conditions, nitrogen and carbon pools are re-initialized following Sitch et al. (2003). Hence, all N from the initialization is
removed that is not supporting plant growth because of other constraints such as water shortage (e.g. in deserts).

A second spin up phase of 390 yrs is conducted for all versions, including land-use change (except in the PNV runs *LPJmL35-PNV* and *LPJmL5-PNV*) by using land-use input of Fader et al. (2010) to capture the influence of historic land-use change on the carbon and nitrogen pools in soil and vegetation.

**4   Results**

Simulations with LPJmL 5 result in carbon pools and net biome productivity (NBP), NPP and GPP fluxes comparable to the carbon-only LPJmL 3.5 version (Table 3) and show a similar temporal dynamic (Fig. 4). The actual vegetation carbon pool is strongly limited by current N levels and increases substantially across all ecosystems, when N limitations are lifted



**Table 3.** Global carbon pools (soil and vegetation carbon) and fluxes (net biome productivity NBP, net primary productivity NPP, and gross primary productivity GPP) for the 6 different experiments (averages over the period 2000 to 2009)

| C pools/fluxes | *LPJmL35* | *LPJmL35-PNV* | *LPJmL5* | *LPJmL5-nL* | *LPJmL5-nF* | *LPJmL5-PNV* |
|---|---|---|---|---|---|---|
| NBP (PgC yr$^{-1}$) | 0.269 | -1.561 | 0.297 | 0.329 | 0.277 | -1.744 |
| NPP (PgC yr$^{-1}$) | 57.12 | 58.90 | 58.85 | 67.71 | 58.24 | 67.74 |
| GPP (PgC yr$^{-1}$) | 129.9 | 143.0 | 134.6 | 153.7 | 133.6 | 168.5 |
| Soil C (PgC) | 2034 | 2156 | 2394 | 2461 | 2389 | 2578 |
| Vegetation C (PgC) | 450.7 | 627.4 | 466.3 | 611.5 | 465.9 | 659.8 |

(*LPJmL5-nL*). Under actual N limitations and static current fertilizer levels (Elliott et al., 2015), global GPP is relatively stable throughout the simulation period (1901-2009, yellow line in Fig. 4a) as the expansion of cropland into increasingly low-input areas compensates the increase in GPP in the natural vegetation (red line in Fig. 4a). NPP increases in the standard simulation with dynamic land use (*LPJmL5*), but not as strongly as for natural vegetation (compare red and yellow lines in Fig. 4b). ,

indicating that the agricultural land is increasingly N limited, so that C:N-ratio dependent maintenance respiration declines and NPP increases whereas GPP does not. The difference in global annual NPP between simulations with natural vegetation only and dynamic land use increases significantly from 3% (*LPJmL35-PNV-LPJmL3.5*) to 16% (*LPJmL5-PNV-LPJmL5*). This indicates that the agricultural land is increasingly N limited, so that C:N-ratio dependent maintenance respiration declines and NPP increases whereas GPP does not. Land-use driven declines in vegetation carbon over the 20th century are similar between

the carbon-only *LPJmL35* and the simulation with nitrogen *LPJmL5* (Fig. 4c), but soil carbon stocks decline with nitrogen, whereas increases in the natural vegetation balance the land-use change induced losses in the carbon-only version (Fig. 4d).

The role of human land use for the limitation of plant growth by nitrogen availability is apparent when comparing simulations with land use (*LPJmL5*, red lines in Fig. 5) and natural vegetation only (*LPJmL-PNV*, orange lines in Fig. 5). The nitrogen pool in the natural vegetation remains quite stable (Fig. 5a) and C:N ratios only slightly increase (Fig. 5b) whereas vegetation

nitrogen with the inclusion of the historical land use declines and N limitation increases by a factor of 2. The predominant difference between both simulations is the 60% increase in losses of N by leaching under land use (Fig. 5c).

The comparison of global N values to literature estimates is divided between values including natural vegetation only and those considering land-use dynamics (Table 4). Whereas several estimates exist for global N pools and fluxes under natural vegetation, those including land-use dynamics are rather rare and given mostly for emissions from the soil (e.g. denitrification

or N$_2$O). Lifting N limitation (*LPJmL5-nL*) results in similar responses for all global nitrogen pools (Table 4). Vegetation N and plant uptake increase substantially by ≈75 and ≈100% respectively, whereas soil N pools only increase slightly by 6%. The omission of N fertilizers has little effects on N pools, which are dominated by the natural ecosystems, but strongly affect nitrogen losses, especially leaching and volatilization fluxes (Table 4). A comparison to literature estimates are discussed further in section 4.1.1.

The approximated relationships between leaf C:N ratios and storage organ C:N ratios based on Bodirsky et al. (2012) lead to consistent but variable C:N ratios in harvested crop organs, reflecting differences between crop types (Fig. 6). The leguminous

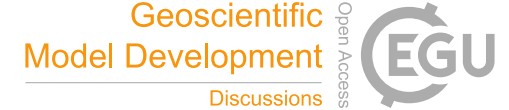



**Table 4.** Global nitrogen pools and fluxes for the 4 different experiments with LPJmL 5 and literature estimates (averages over the period 2000 to 2009)

| N pools/fluxes | LPJmL5 | Literature estimates LU | LPJmL5-PNV | Literature estimates PNV | LPJmL5-nL | LPJmL5-nF |
|---|---|---|---|---|---|---|
| Vegetation (PgN) | 2.29 | - | 3.21 | $3.6^1$, $3.8^2$, $5.3^3$, $16^4$ | 4.00 | 2.28 |
| Soil organic matter (PgN) | 130.9 | - | 138.9 | $120^1$, $101^2$, $61.4^3$, $280^4$, $95^5$ | 138.8 | 130.6 |
| Soil $NH_4^+$ (TgN) | 640.7 | - | 650.7 | $361^3$ | - | 637.7 |
| Soil $NO_3^-$ (TgN) | 5873 | - | 6282 | $580^3$ | - | 5636 |
| Plant uptake (TgN yr$^{-1}$) | 751 | - | 836 | $970^1$, $1130^2$, $1080^3$, $620^4$ | 1526 | 718 |
| Mineralization (TgN yr$^{-1}$) | 1422 | - | 1715 | $980^1$, $1030^2$, $6300^4$ | 1766 | 1404 |
| Immobilization (TgN yr$^{-1}$) | 781.6 | - | 937.3 | - | 774.7 | 778.2 |
| Leaching (TgN yr$^{-1}$) | 72.63 | $93^6$, $95^7$ | 45.39 | $13^1$, $87^2$, $5^4$ | - | 46.17 |
| Volatilization (TgN yr$^{-1}$) | 19.11 | - | 15.27 | - | - | 14.03 |
| Denitrification $N_2O$ emissions (TgN yr$^{-1}$) | 6.43 | - | 5.89 | - | - | 5.67 |
| Denitrification $N_2$ emissions (TgN yr$^{-1}$) | 52.05 | $68^6$ | 47.63 | - | - | 45.88 |
| Denitrification total (TgN yr$^{-1}$) | 58.48 | $72-85^6$, $25^7$, $67^9$ | 53.52 | - | - | 51.55 |
| Nitrification $N_2O$ (TgN yr$^{-1}$) | 7.37 | - | 8.33 | - | - | 6.74 |
| total $N_2O$ emissions (TgN yr$^{-1}$) | 13.80 | $11^8$, $15^9$ | 12.63 | - | - | 12.41 |
| Biological N fixation (TgN yr$^{-1}$) | 134.4 | $92^6$, $118^7$, $104-108^8$, $107^9$ | 102.9 | $34^1$, $108^2$, $211^4$, $58^{10}$ | 136.3 | 134.3 |

[1] Smith et al. (2014), [2] Zaehle et al. (2010a), [3] Xu-Ri and Prentice (2008), [4] Lin et al. (2000), [5] Post et al. (1985), [6] Bouwman et al. (2013), [7] Sutton et al. (2013), [8] Galloway et al. (2013), [9] Galloway et al. (2004), [10] Vitousek et al. (2013)





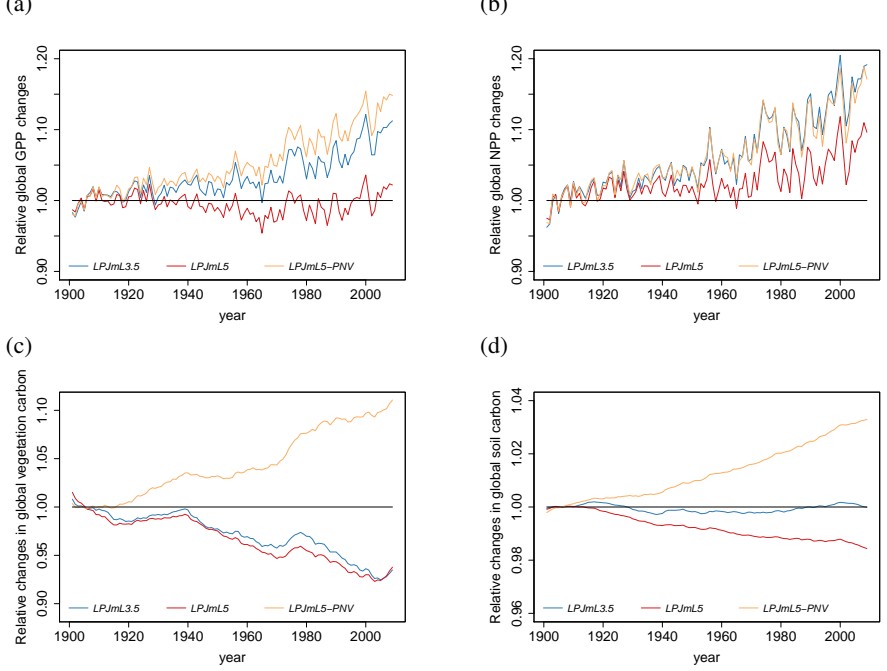

**Figure 4.** Relative global changes of GPP (a), NPP (b), vegetation carbon (c) and soil carbon (d). Values are normalized to the 1901-1910 average to make the different model versions and settings comparable.

crops (soybean, pulses) are not limited by N, as they can acquire the necessary N via biological N fixation. For these, C:N ratios of harvested organs are typically underestimated. Under unlimited N supply, C:N ratios are typically reduced (Fig. 6b).

We find that agricultural land use and associated fertilizer application greatly increases nitrogen pollution. Leaching (+60%) and ammonia volatilization (+25%) increase strongly, which is almost entirely driven by fertilizer inputs, not by land-use

change (compare *LPJmL5* with *LPJmL5-PNV* and *LPJmL5-nF* in Table 4). On the contrary, $N_2O$ emissions only change slightly, when agricultural land use is accounted for as increases in denitrification are partially compensated by decreases in nitrification under reduced net mineralization (mineralization minus immobilization flux) of soil organic matter (Table 4). The effect of agricultural land use and fertilizer application is also clearly detectable in the spatial patterns of leaching. The ratio of *LPJmL5-PNV* to *LPJmL5* (Fig. 7a) is mostly below 1 indicating higher leaching in agricultural areas. In natural vegetation

under dry conditions also ratios above 1 can occur (Fig. 7b).

When N limitations are lifted through unlimited N supply (*LPJmL5-nL*), GPP is mostly increased, except in very dry environments. The scatter plot (Fig. 8b) shows that the GPP increase mainly occurs in low to moderately productive areas. Decreases in GPP under unlimited N supply are possible where other factors are strongly limiting (e.g. water) and the higher N supply leads to higher maintenance respiration under lower tissue C:N ratios, so that less biomass is available for leaves and

thus less light can be intercepted.





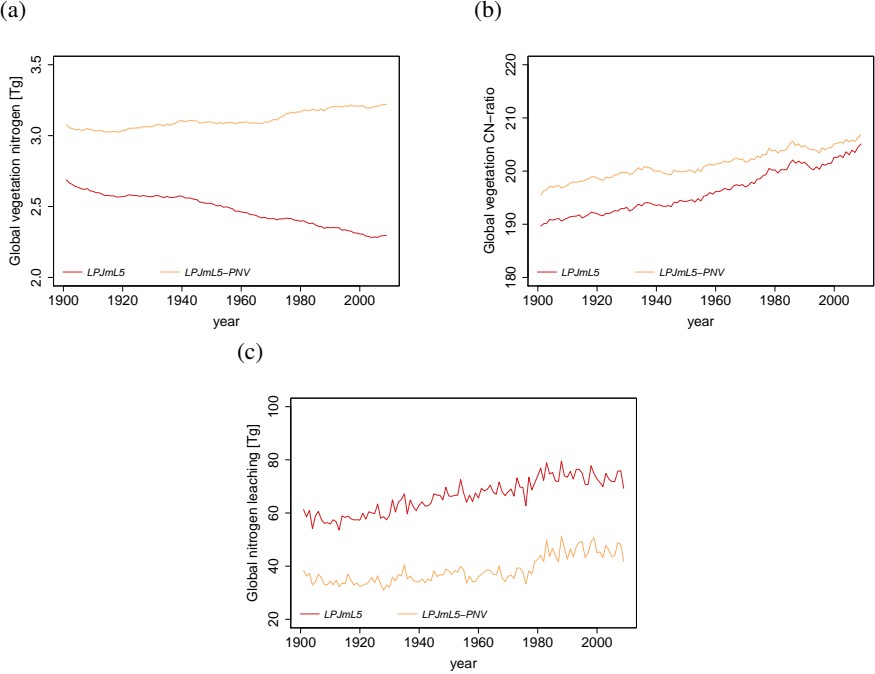

**Figure 5.** Simulated global time series of vegetation nitrogen (a), vegetation C:N ratios (b) and leaching (c) with land use (*LPJmL5*) and potential natural vegetation (*LPJmL5-PNV*).

## 4.1 Model evaluation

To evaluate model performance, we here focus on carbon and nitrogen pools and fluxes at global and specific sites. Many estimates are also model-based, so that these cannot be used for model evaluation but only for putting our simulation results into context.

### 4.1.1 N pools and fluxes

Typically, simulated N pools and fluxes are within literature estimates (Table 4), although literature estimates are often broad, reflecting substantial uncertainty in these values. Vegetation N of the potential natural vegetation simulation, *LPJmL5-PNV*, is slightly below the other model-based estimates (Table 4), whereas other fluxes (e.g. plant uptake of N or mineralization) are within the range of reported values. For simulations with land use history, *LPJmL5*, comparison is possible for most of the emissions from the soil. There our values for leaching and $N_2$ emissions are slightly below other estimates. For the complimentary flux, $N_2O$ emissions from denitrification, there is no other estimate, but total $N_2O$ emissions from denitrification and nitrification are within the range of other estimates again (Table 4). Xu-Ri and Prentice (2008) are the only study reporting global soil pools of mineral N forms, but for potential natural vegetation only and for the upper 1.5 m soil layer. Our values, which are the inventory of 3 m soil, are much higher than the estimates of Xu-Ri and Prentice (2008) by a factor of 2 and 10,




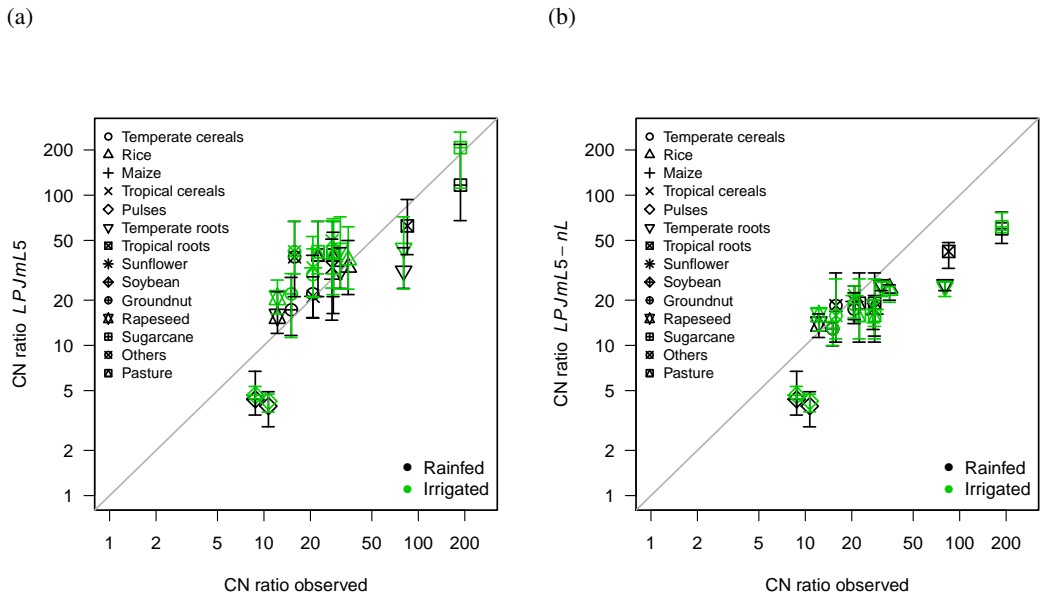

**Figure 6.** Observed C:N ratios of harvested crops versus simulated mean ratios for the crop PFTs (a) with N limitation (*LPJmL5*) and (b) without N limitation (*LPJmL5-nL*). The vertical error bars denote the 95% percentile.

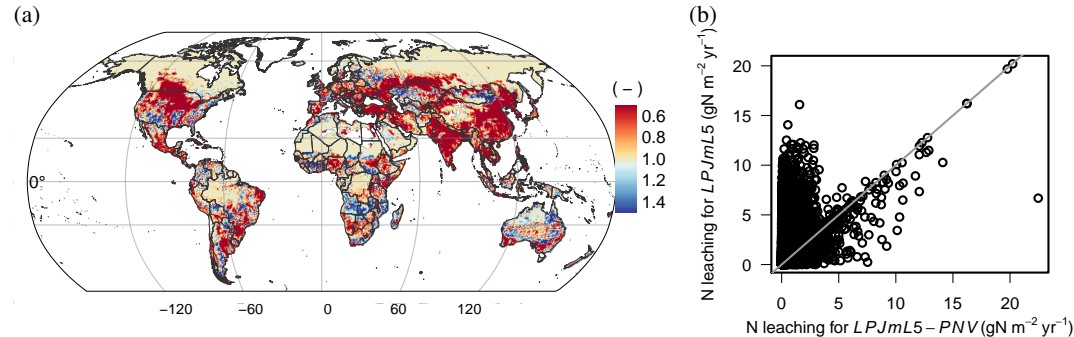

**Figure 7.** Ratio of leaching flux under potential natural vegetation (*LPJmL5-PNV*) to the leaching flux under actual land-use patterns (*LPJmL5*) (a); values less than 1 indicate higher leaching under actual land-use patterns. The scatter plot (b) shows that leaching is increased strongly mostly in regions where leaching is low under potential natural vegetation.

respectively, for soil $NH_4^+$ and $NO_3^-$. Accounting for the content of the first 1.5 m, soil $NO_3^-$ amounts to 847 TgN and soil $NH_4^+$ to 228 TgN. Therefore, in comparison to Xu-Ri and Prentice (2008) we overestimate $NO_3^-$ values and underestimate $NH_4^+$ values in the soil but are in the same order of magnitude.





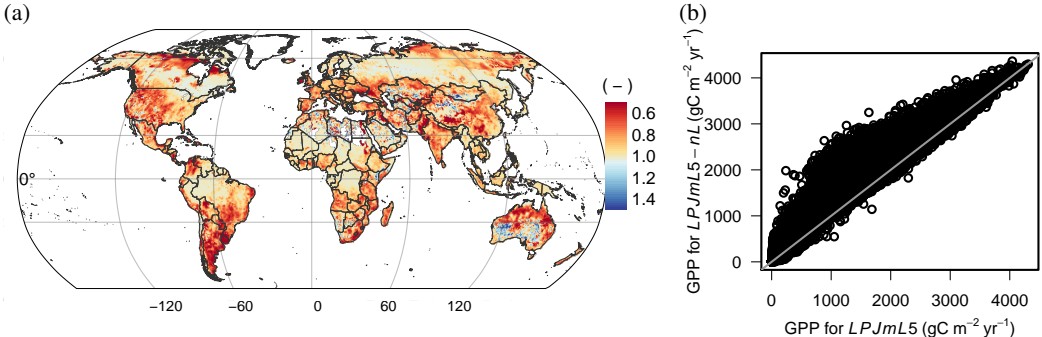

**Figure 8.** Ratio of GPP under actual N limitations (*LPJmL5*) to unlimited N supply (*LPJmL5-nL*) (a); values less than 1 indicate higher GPP under unlimited N supply. The scatter plot (b) shows that GPP is increased through additional N supply mostly in low to moderately productive regions.

### 4.1.2 Carbon cycle dynamics

Carbon dynamics are mostly unchanged to the predecessor version LPJmL 3.5 (Table 3 and Fig. 4). In comparison to measured site-level GPP, NPP and vegetation carbon, LPJmL 5 performs well, especially for GPP and NPP, but with a tendency to underestimate vegetation carbon (Fig. 9). We also provide comparisons to eddy flux tower measurements (ORNL DAAC, 2011) in the supplement. SI Figs. S5-S11 show modelled versus observed net ecosystem exchange (NEE) rate defined as

$$NEE = NPP - R_h, \tag{70}$$

where $R_h$ is the heterotrophic respiration. For some sites a time lag (e.g. site Renon/Ritten) between modelled and observed is visible. Because LPJmL 5 uses the phenology scheme of LPJmL 3.5 incorporating the new phenology of LPJmL 4 might reduce these deviations. The overall agreement between modelled and observed NEE is satisfying. Also the simulated evapo-transpiration fluxes shown in SI Figs. S12-S20 agree very well with the observations.

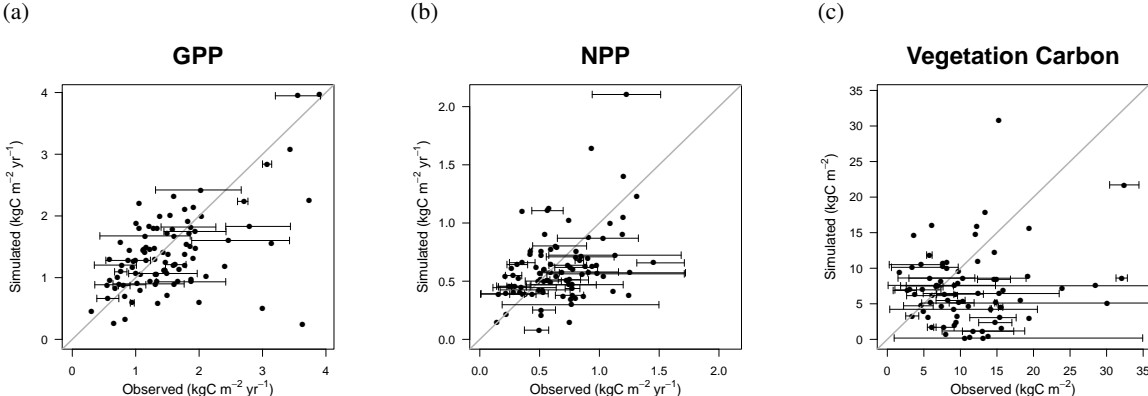

**Figure 9.** Observed versus simulated GPP (a), NPP (b) and vegetation carbon (c).

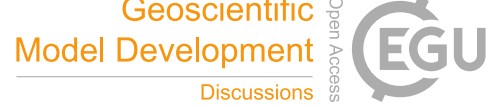



### 4.1.3 Crop yields

The implementation of nitrogen limitation also greatly helps to improve the simulation of global patterns of crop productivity. Regional differences in crop productivity, which had to be calibrated via a scaling of the maximum leaf area index ($LAI_{max}$), as described in Fader et al. (2010), can now largely be reproduced by the model, except for regions where fertilizer inputs and

climate conditions are not the only yield limiting factors. The temporal variance of simulated crop yields is often not affected much by accounting for N limitation and sometimes improves or worsens the time series correlation with FAO yield statistics (FAOSTAT), (Fig. 10 and SI Figs. S21–S23 in the supplementary material).

We also use the online tool as supplied by Müller et al. (2017) for comparing the crop yield simulations against the Global Gridded Crop Model Intercomparison (GGCMI) model ensemble. Also here, results show that LPJmL 5 improves with re-

spect to reproducing absolute yield levels across different countries, but that there is little effect on the simulated inter-annual variability of crop yields. As with the calibrated LPJmL 3.5 version (Fader et al., 2010), the uncalibrated LPJmL 5 simulations perform well in comparison to the other GGCMI models. We supply the output of that online model evaluation tool as supplementary material.

## 5   Discussion and conclusions

The current implementation of nitrogen dynamics into LPJmL 3.5 forming LPJmL 5 introduces a missing feature into a unique modeling framework of the terrestrial biosphere. LPJmL 5 combines natural vegetation dynamics, the full terrestrial hydrology and managed grass- and croplands in one consistent framework with the associated carbon, water, and now also nitrogen pools and fluxes. Owing to parallel model development efforts, LPJmL 5 does not yet include all model features of the first open source version of LPJmL, LPJmL 4 (Schaphoff et al., 2017b, a), such as the updated allocation scheme for managed grasslands

(Rolinski et al., 2017) and the updated phenology scheme for natural vegetation (Forkel et al., 2014).

With the implementation of the nitrogen dynamics, the model simulations require new inputs, especially on atmospheric deposition, but also on fertilizer applications, where we currently use a static crop- and irrigation-specific data set, developed for the harmonization of crop models in the Agricultural Model Intercomparison and Improvement Project (AgMIP) (Elliott et al., 2015). This static fertilizer set also affects the simulation of historic carbon cycle dynamics, as high-input regions, such

as large parts of Europe or northern America receive current high N inputs also in the early 20th century, whereas historic land expansion mostly moves into regions with currently lower input systems. As a consequence, land-use change leads to increasing nitrogen limitation and increasing C:N ratios, which may be an artifact from the static fertilizer input data set used.

As the historic land-use development and fertilizer application are important for simulated current biogeochemical cycles, historic time series of crop-specific fertilizer application would be desirable. Also, global data sets on crop rotations (Kollas

et al., 2015), timing of field operations (Hutchings et al., 2012) or crop residue management, as well as livestock management systems (Rolinski et al., 2017) would be an asset, as the interaction of different cropping systems and natural vegetation is now further increased via the nitrogen cycle.





**Figure 10.** Maize yield simulations (in tFM ha$^{-1}$) for the 10 top-producing countries for the carbon-only LPJmL 3.5 version, the version with N limitation and with unlimited N supply.



This first implementation of nitrogen dynamics into LPJmL constitutes an operational modeling framework with many detailed processes resolved explicitly. Specific processes are currently implemented in a simplified manner, even though more detailed approaches are available, as, e.g., for biological N fixation (Fisher et al., 2010). As such, some process may have to be revised upon further testing against new reference data. Additional reference data would greatly help to evaluate model

performance, which currently is largely constrained to comparisons against other modeling results.

LPJmL 5 constitutes a unique modeling framework that can now simulate global terrestrial carbon, water and nitrogen dynamics, consistently accounting for natural vegetation dynamics, agricultural crop- and grassland management and water management.

*Code and data availability.* The source code is available upon request from the main author for the review process and for selected collabo-

rative projects. We are in the process of setting up a publicly accessible open-source model code repository where the code as described here will be the released. The source code will be generally available after final publication of this paper and a DOI for access will be provided. Data from the simulations conducted here is available upon request from the main author.

*Competing interests.* The authors declare that they have no conflict of interest.

*Acknowledgements.* This study was supported by the German Federal Ministry of Education and Research's (BMBF's) "PalMod 2.3 Methankreis-

lauf, Teilprojekt 2 Modellierung der Methanemissionen von Feucht- und Permafrostgebieten mit Hilfe von LPJmL", (FKZ 01LP1507C). C.M. and S.R. acknowledge financial support from the MACMIT project (01LN1317A) funded through the German Federal Ministry of Education and Research (BMBF). S.Z. was supported by the European Research Council (ERC) under the European Union's Horizon 2020 research and innovation programme (QUINCY; grant no. 647204).



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
