# Peer review of "Implementing the Nitrogen cycle into the dynamic global vegetation, hydrology and crop growth model LPJmL (version 5)"

_Geoscientific Model Development, 2017_

## Editor Comment (EC1) · J. C. Hargreaves (Editor) · 18 Oct 2017

The authors have sent me a copy of the code, so reviewers who wish to remain anonymous may request the code from me via email.

---

## Author Comment (AC1) · 19 Oct 2017

[Figure]

**Figure 10.** Maize yield simulations (in tFM ha$^{-1}$) for the 10 top-producing countries for the carbon-only LPJmL 3.5 version, the version with N limitation and with unlimited N supply.

[Figure]

**Figure S21.** Wheat yield simulations (in tFM ha$^{-1}$) for the 10 top-producing countries for the carbon-only LPJmL 3.5 version, the version with N limitation and with unlimited N supply.

[Figure]

**Figure S22.** Rice yield simulations (in tFM ha$^{-1}$) for the 10 top-producing countries for the carbon-only LPJmL 3.5 version, the version with N limitation and with unlimited N supply.

[Figure]

**Figure S23.** Soybean yield simulations (in tFM ha$^{-1}$) for the 10 top-producing countries for the carbon-only LPJmL 3.5 version, the version with N limitation and with unlimited N supply.

---

## Referee Comment (RC1) · Anonymous Referee #1 · 6 Jan 2018

**Overview**

von Bloh and co-authors present a N enabled version of the LPJmL model that couples terrestrial biogeochemistry and biogeophysics in a dynamic vegetation model. They evaluate global (and regional) changes in terrestrial C and N dynamics over the 20th century and compare their findings with other literature estimates. The work marks an important development for the community of LPJmL users, and I encourage the authors to revise their manuscript to make it more relevant for a broader readership interested in representing global biogeochemistry in land models.

**Major concerns**

I appreciate that this is largely a model development and documentation paper, but found some of the discussion surrounding main display items rather hasty and lacking appropriate depth. For example, the spatial patterns of N limitation (Fig. 8) suggest that both tropical and boreal forests are not N limited (indeed, forests globally may have a low sensitivity to N availability)? This is just one example, but on revisions I would encourage the authors to unpack and explore their findings a bit more. Take the opportunity to call out strengths and weaknesses in the current approach and discuss particular model assumptions that are responsible for these features. This doesn't have to be exhaustive, but will help add depth to the results and discussion.

Page 1, Line 6. Significant improvements in crop yields are not apparent in the updated Fig. 10 and SI material uploaded by the authors. Would a plot of global crop yields vs. observations make this point more clearly? If crop model improvements are the big advancement in the current model development, I'd encourage more attention be given to establishing these improvements in the main text. That said, the estimates of N losses (through leaching and N2O emissions) also have important regional and global consequences and seem to be done well in this version of the model. Should these accomplishments be highlighted in the abstract too?

Minor comments and technical concerns

P1, L 8. This suggest the is still scaling occurring in regions with favorable climate and N inputs? Is this true?

P1, L 18 Zaehle and co-authors (2015) made similar findings, seems worth citing here?

Throughout section 2 is it worth briefly distinguishing the similarities and differences in the approach taken with LPJmL vs. other land models, especially LPJguess (Smith et al 2014) which is referenced throughout?

P4, L 6. It seems odd to introduce table 2 before table 1. Similarly, table 4 is introduced before table 3 (page 7).

[Figure]

Section 2.2. Is the soil biogeochemistry for this version of the model vertically resolved, as implied with e.q. 7? If so should this be mentioned in section 2: model description?

Eq. 13-14, could pft specific root distribution parameters be easily described in the current tables or included elsewhere?

P5, L 27. As written this sounds like rates of GPP are reduced by respiration rates? This strikes me as strange. Wouldn't autotrophic respiration be subtracted from GPP to calculate NPP rates (eq. 18). This also isn't clear in Fig 1.

Figure 3: check to see the colors for each arrow are labeled correctly and/or defined. Alternatively, the approach here seems pretty standard, I wonder if the distracting rainbow of flux arrows are really necessary?

Section 2.6 How are all the competing fates of inorganic N solved (e.g. sequentially, simultaneously, etc)?

Section 2.6. Are litter and SOM pools have a fixed C:N ratio or are they flexible (w/in bounds).

P 15 L 23 What is a " nodulating leguminous crops pulses", the phrase seems redundant? Maybe just use "soybeans and pulses".

P17, line 13. What happens to the other 40% of the manure? Is it not really applied, or does it go into SON pools?

Section 3. What spatial resolution are these simulations? Does each grid cell have a single pft, or is there subgrid variability of vegetation? If there is subgrid PFT variation, do all plants share a soil column, or to they each have individual columns (that is, does manure and fertilizer applied stay on crop only soils, or is it available to plants throughout the grid)?

Table 4: Although they are described in the main text (section 3) the abbreviations for experiments used in column headings are non-intuitive enough to prevent the information contained in the display item from standing on its own. Consider adding text to the the table heading or columns to make these data more understandable.

Table 4. I'm surprised global NO3 pools are an order of magnitude larger than the NH4 pools. I wonder if there is spatial structure to these patterns (e.g. accumulation of NO3 is warm or arid regions), or if the patterns is relatively globally distributed. Regardless, it seems surprising given the relatively high mobility and multiple loss pathways of NO3, compared with NH4, and suggests that nitrification rates may be too high in the model? Alternatively, decomposition rates may be high, supplying excess inorganic N, or plant NO3 uptake may be underestimated? This may be worth mentioning in the discussion (section 4.1.1 or 5)?

P 18 L 8 Why are agricultural lands (that are presumably being fertilized) becoming increasingly N limited? Is there some metric of N limitation that can illustrate this point more directly, as it's not intuitive from Fig 4a,b. Also, it seems odd to increase carbon use efficiency (NPP:GPP) if the system is becoming more N limited? I see how it occurs in the model, because of higher tissue C:N ration and lower RA costs, but is it ecologically realistic?

Fig 4. How were relative GPP changes calculated, I didn't see this described in the text? Also, consider adding information about line colors to the figure caption, as the legend insets are very small and hard to read.

Fig. 4 It looks like the two control models (3.5 and 5) lose vegetation and soil C throughout the 20th century, but GCP data suggests the land surface should be a C sink, at least over then end of the 20th century (e.g., LeQuere et al, 2015). Given increases in plant productivity in Fig 4a,b- this suggests the land use C change flux must be pretty large?

P 18 L 15. Where is the data showing that N limitation increases by a factor of two? How is N limitation being assessed in this statement?

P 18 L 16. What's causing the higher leaching losses with the control model? Does it have to do with vegetation demand for N, rooting profiles of managed vegetation, or other factors?

Table 4. Its it worth discussing limitations (or uncertainties) of some of the 'observational' estimates presented here

Fig 6. Agreement on crop C:N ratios doesn't seem that surprising, given the ranges for leaf C:N and allocation that are proscribed in the model (Tables 1,2) . I'm assuming the values for R3 in Table 2 were tuned to provide the spread shown in Fig 6? This is fine, but should be acknowledged. In the text.

Fig 7a, it strikes me as odd to have low values (<1) indicative of high leaching losses, especially when points above the 1:1 line show areas of high leaching under present vegetation (fig 7b)

Fig 9, where do the obs come from- especially for NPP and Veg C. Is each point supposed to represent an individual sites? This are from the same FluxNet sites as in the SI figures? Also, where do the observational error bars come from & how were they calculated? Finally, should correlation coefficients & significance be reported?

Section 4.1.3 Where are the LAI data shown that the addition of N biogeochemistry supposedly fixes? If this is the big advancement with the model presented here should these data also be shown? Is it just the addition of N biogeochemistry that's responsible for the proported improvements, or were other parametric or structural changes made?

Fig. 10 A bunch of questions: What are the residuals and how are they calculated? What are units for the y-axis (what is the 'FM')? What are the little numbers in the top of each panel showing? Finally, it looks like global maize production increases over the period shown but the models are all flat. What's driving the increase in yields that the model is apparently missing? Is this true for other major crops?

P 26 L 4, What improvements are necessary? What additional complexity may improve

things further? What data are critical to getting terrestrial C-N dynamics less wrong? As presented these are kind of empty / throwaway statements. Can the be flushed out with some more detail, both in the main text and in this summary conclusion?

References: Le Quéré C, et al. (2015) Global Carbon Budget 2015. Earth System Science Data 7: 349-396. DOI:10.5194/essd-7-349-2015

Smith et al. (2014) Implications of incorporating N cycling and N limitations on primary production in an individual-based dynamic vegetation model, Biogeosciences, 11, 2027-2054, doi:10.5194/bg-11-2027-2014.

Zaehle, et al. (2015). Nitrogen availability reduces CMIP5 projections of twenty-first-century land carbon uptake. Journal of Climate 28.6: 2494-2511.

---

## Referee Comment (RC2) · Anonymous Referee #2 · 13 Jan 2018

General comments:

This study presents the development of a C-N coupled DGVM by incorporating Nitrogen cycle into a carbon-only version of LPJmL. DGVMs with representation of coupled C-N cycles could help better understand the terrestrial greenhouse gas fluxes (e.g., future carbon sink capacity). The authors then evaluate model performance of the global carbon and nitrogen pools/fluxes against literature values. A specific effort was made on evaluating the performance of simulating crop productivity. The model development is clearly described, while the model evaluation part needs some more efforts on the organization, clarification, and deeper investigation of results.

[Figure]

Specific comments:

1. The results part before section 4.1 presents the comparison between outputs from global simulations with various model versions and setups with some interesting findings. However, the section is hard to follow without clear organization of the results, and some interesting findings and important characteristics are presented without proper investigations/explanations. I would suggest authors to make it a sub-section with clearer titles/points indicating the major content of each paragraph. A little deeper investigations or more detailed explanations on the differences between versions/setups would be preferable. In addition, model versions/setups should always be presented along with results (sometimes, it is missing).

2. Section 4.1.1 and 4.1.2 present the comparison of N and C pools/fluxes at both site-level and globally against literature values. Besides comparing values, some more explanations/discussions in the differences/discrepancies could be helpful for readers to understand the model performance.

3. The authors present the improvement in simulating crop productivity by model developed in this study very briefly in Section 4.1.3, while it was shown as major results consisting almost half of the Abstract, which might make readers think the evaluation is only done for crop productivity. Even it is the major part of evaluation, I actually did not get the 'improvements' that authors claimed from the text. Though model results and its comparison with data were shown in Fig. 10 and lots of SI Figures, readers can only get little information on the model performance and the improvement due to implementation of N cycle (at least for me). With comparison on many sites, no statistically synthesis of the performance (no numbers), and no further investigations on 'how and why' were shown in the text. It is necessary/critical to show quantitatively how the improvement can be supported by the results. It is not the job of readers to check every figure, compare the lines, and draw the conclusion. In addition, there is no line for LPJmL3.5 in SI Fig. S5-S20, how the 'improvements' can be seen?

[Figure]

4. The authors tried to discuss the improvements and limitations of this study in 'Discussion and conclusions' section. But it is lack of clear structure. A synthesis of limitation might be helpful (e.g., listing limitations points by points at first).

Minor comments (there might be some duplicate points as some listed above):

P18L1-3: Need to indicate the results come from which version/setup. The means of red/yellow lines were opposite in the text and the figure. Fig. 4: It is necessary to show LPJmL35-PNV too. All lines in the figures of this manuscript should be drawn with thicker lines. P18L14: It seems not 'quite stable' and not 'slightly increase' in Fig. 5. And how the factor of 2 be derived? P18L22: Please give model setup. P20L9-10: What ratio? It is not clear. It would be necessary to explain why the ratios above 1 can occur. P21L9-13: I don't understand these 3 sentences. What's the meaning? Figure 9: Please indicate the data sources for the observed values. P23L5: It is necessary to show in SI figures the LPJmL3.5 output for a real comparison. P23L9: Please indicate what is 'satisfying', is there any indicators to draw this conclusion? P24L5: Please indicate where the climate conditions are not the only yield limiting factors. Sect. 4.1.3: Please indicate how well the model performance. We can get nothing solid from the text.

---

## Author Response (AR1)

**Response to reviewer #1**

Major concerns:

I appreciate that this is largely a model development and documentation paper, but found some of the discussion surrounding main display items rather hasty and lacking appropriate depth. For example, the spatial patterns of N limitation (Fig. 8) suggest that both tropical and boreal forests are not N limited (indeed, forests globally may have a low sensitivity to N availability)? This is just one example, but on revisions I would encourage the authors to unpack and explore their findings a bit more. Take the opportunity to call out strengths and weaknesses in the current approach and discuss particular model assumptions that are responsible for these features. This doesn't have to be exhaustive, but will help add depth to the results and discussion.

Page 1, Line 6. Significant improvements in crop yields are not apparent in the updated Fig. 10 and SI material uploaded by the authors. Would a plot of global crop yields vs. observations make this point more clearly? If crop model improvements are the big advancement in the current model development, I'd encourage more attention be given to establishing these improvements in the main text. That said, the estimates of N losses (through leaching and N2O emissions) also have important regional and global consequences and seem to be done well in this version of the model. Should these accomplishments be highlighted in the abstract too?

**Answer: The results section has been expanded. We have now structured results into 3 sections: Carbon pools and fluxes, Nitrogen pools and fluxes, Land use and nitrogen dynamics. In particular, Taylor diagrams of regional crop productivities have been added to the manuscript and supplement. They clearly indicate that significant parts of the calibration previously needed are now no longer necessary, as N limitation already reduces simulated crop yields in low-input countries. The strong influence of land use on nitrogen losses is now mentioned in the abstract.**

Minor comments

P1, L 8. This suggest the is still scaling occurring in regions with favorable climate and N inputs? Is this true?

**Answer: Scaling of yields has still to be done for regions where climate and nitrogen supply are not the main limiting factor, but, e.g. phosphorus, poor pest management or other management aspects not explicitly considered in the model as now mentioned in section 4.4.3.**

P1, L 18 Zaehle and co-authors (2015) made similar findings, seems worth citing here? Throughout section 2 is it worth briefly distinguishing the similarities and differences in the approach taken with LPJmL vs. other land models, especially LPJguess (Smith et al 2014) which is referenced throughout?

**Answer: In fact, nitrogen dynamics have been incorporated in many other dynamical vegetation models (as described on P1, L22). As LPJ-GUESS, our model considers not only natural vegetation but**

**also takes into account managed crops. In contrast to LPJ-GUESS nitrogen transformation in soils are simulated in a more sophisticated way in LPJmL5, including immobilization of nitrogen. This is now specified in the introduction and model description.**

P4, L 6. It seems odd to introduce table 2 before table 1. Similarly, table 4 is introduced before table 3 (page 7).

**Answer: The order of tables has been changed following the suggestions of the referee.**

Section 2.2. Is the soil biogeochemistry for this version of the model vertically resolved, as implied with e.q. 7? If so should this be mentioned in section 2: model description?

**Answer: Soil processes are vertically resolved in 6 soil layers. A sentence has been added to the model description.**

Eq. 13-14, could pft specific root distribution parameters be easily described in the current tables or included elsewhere?

**Answer: The parameterization of PFT-specific root distribution is now described in the paper and the corresponding parameters included in Table 2.**

P5, L 27. As written this sounds like rates of GPP are reduced by respiration rates? This strikes me as strange. Wouldn't autotrophic respiration be subtracted from GPP to calculate NPP rates (eq. 18). This also isn't clear in Fig 1

**Answer: Indeed, for calculating NPP, respiration has to be subtracted from GPP but GPP itself is not directly reduced. This has been clarified in section 2.4. In figure 1, GPP (flux) is not shown, but the process (photosynthesis).**

Figure 3: check to see the colors for each arrow are labeled correctly and/or defined. Alternatively, the approach here seems pretty standard, I wonder if the distracting rainbow of flux arrows are really necessary?

**Answer: Indeed, the approach is based on standard implementations. We find a colored scheme more helpful than a black-and-white figure but acknowledge that others may have other color preferences. The boxes, however, are now colored in black.**

Section 2.6 How are all the competing fates of inorganic N solved (e.g. sequentially, simultaneously, etc)?

**Answer: The transformation of nitrogen is calculated in sequential order as now stated in section 2.6.**

Section 2.6. Are litter and SOM pools have a fixed C:N ratio or are they flexible (w/in bounds).

**Answer: Litter and SOM pools have flexible C:N ratios now explicitly stated in section 2.6.**

P 15 L 23 What is a " nodulating leguminous crops pulses", the phrase seems redundant? Maybe just use "soybeans and pulses".

**Answer: There are non-nodulating legumes (doi:10.14719/pst.2015.2.2.97) as well as non-nodulating soybean varieties (doi::10.1007/978-94-009-4482-4_19) and we wanted to make clear that we're considering these leguminous crops as nodulating varieties with capacities to fix nitrogen.**

P17, line 13. What happens to the other 40% of the manure? Is it not really applied, or does it go into SON pools?

**Answer: No, the data from Elliott et al. (2015) only provide mineral reactive N forms. In their data set, only the plant available fraction (60% of manure) is specified, the rest is ignored. This is now clarified in the text.**

Section 3. What spatial resolution are these simulations? Does each grid cell have a single pft, or is there subgrid variability of vegetation? If there is subgrid PFT variation, do all plants share a soil column, or to they each have individual columns (that is, does manure and fertilizer applied stay on crop only soils, or is it available to plants throughout the grid)?

**Answer: Simulations are carried out for a resolution of 0.5 x 0.5°. PFTs can be established concurrently within a cell competing for light, water and nitrogen sharing the same soil resources. This has been explained in more detail in the paper.**

Table 4: Although they are described in the main text (section 3) the abbreviations for experiments used in column headings are non-intuitive enough to prevent the information contained in the display item from standing on its own. Consider adding text to the table heading or columns to make these data more understandable.

**Answer: The table captions now include an explanation of the different suffixes for the abbreviations of the experiments.**

Table 4. I'm surprised global NO3 pools are an order of magnitude larger than the NH4 pools. I wonder if there is spatial structure to these patterns (e.g. accumulation of NO3 is warm or arid regions), or if the patterns is relatively globally distributed. Regardless, it seems surprising given the relatively high mobility and multiple loss pathways of NO3, compared with NH4, and suggests that nitrification rates may be too high in the model? Alternatively, decomposition rates may be high, supplying excess inorganic N, or plant NO3 uptake may be underestimated? This may be worth mentioning in the discussion (section 4.1.1 or 5)?

**Answer: This is indeed discussed in section 4.1.1 and we have now expanded this discussion. NH4 is constantly converted to NO3 by nitrification, a microbe-mediated process under aerobic conditions, which are often prevailing. NO3 is lost via leaching or denitrification, again a microbe-mediated process but under anaerobic conditions, which are not dominant conditions in most soils. As a consequence, NO3 concentrations are typically much higher than NH4 concentrations, unless N is limiting plant uptake, see e.g. Kabala et al. (2017). Also, NO3 is accumulated in lower soil layers, despite leaching losses. This phenomenon is reported by Walvoord et al. (2003) and Ascott et al.**

**(2017) and reproduced by the model. Still, this nitrogen is mostly unavailable to plants, as these have little root access to the lower soil layers at 3m depth. This is now discussed in the paper.**

P 18 L 8 Why are agricultural lands (that are presumably being fertilized) becoming increasingly N limited? Is there some metric of N limitation that can illustrate this point more directly, as it's not intuitive from Fig 4a,b. Also, it seems odd to increase carbon use efficiency (NPP:GPP) if the system is becoming more N limited? I see how it occurs in the model, because of higher tissue C:N ration and lower RA costs, but is it ecologically realistic?

**Answer: In the simulations, plant growth is enhanced through CO2 fertilization, whereas fertilizer inputs are static. Also, cropland expansion over the 20th century dominantly expands in areas where fertilizer inputs per hectare are low. As such, the global average fertilizer amount (kg N per hectare) declines. This is now also briefly described in the text (section 4.4.3).**

**The relationship of RA costs on C:N ratios is well established and was already implemented in the carbon-only model version. There plant organs had a static prescribed C:N ratio that determined respiration costs for maintenance.**

Fig 4. How were relative GPP changes calculated, I didn't see this described in the text? Also, consider adding information about line colors to the figure caption, as the legend insets are very small and hard to read.

**Answer: Relative changes are calculated by dividing the values by their 1901-1910 average as now stated in the figure caption. The meaning of the different colors in the figures is now explained in the figure caption.**

Fig. 4 It looks like the two control models (3.5 and 5) lose vegetation and soil C throughout the 20th century, but GCP data suggests the land surface should be a C sink, at least over then end of the 20th century (e.g., LeQuere et al, 2015). Given increases in plant productivity in Fig 4a,b- this suggests the land use C change flux must be pretty large?

**Answer: Yes, see also Table 3 where the NBP (net biome productivity) is listed. For natural vegetation (LPJmL3.5_PNV and LPJmL5_PNV), the terrestrial biosphere is a net carbon sink of about 1.5 and 1.7 PgC/yr, whereas when accounting for current land-use patterns (LPJmL3.5 and LPJmL5), the terrestrial biosphere is a net source.**

P 18 L 15. Where is the data showing that N limitation increases by a factor of two? How is N limitation being assessed in this statement?

**Answer: In fact, the statement is misleading and has been omitted from the manuscript.**

P 18 L 16. What's causing the higher leaching losses with the control model? Does it have to do with vegetation demand for N, rooting profiles of managed vegetation, or other factors?

**Answer: The higher leaching under land use is caused by the additional fertilizer input and the irrigation water application under land use, which do not occur in natural vegetation. This is now explained in the text (section 4.3).**

Table 4. It's it worth discussing limitations (or uncertainties) of some of the 'observational' estimates presented here

**Answer: This discussion is placed in section 4.1.1 and we have extended this now. Certainly, estimates based on only one other model have no value for a model evaluation and only serve as a point of reference.**

Fig 6. Agreement on crop C:N ratios doesn't seem that surprising, given the ranges for leaf C:N and allocation that are proscribed in the model (Tables 1,2) . I'm assuming the values for R3 in Table 2 were tuned to provide the spread shown in Fig 6? This is fine, but should be acknowledged. In the text.

**Answer: In fact, R3 values were calibrated to the observed values. The approximated relationships between leaf C:N ratios and storage organ C:N ratios based on Bodirsky (2012} lead to consistent but variable C:N ratios in harvested crop organs, as stated in the manuscript.**

Fig 7a, it strikes me as odd to have low values (<1) indicative of high leaching losses, especially when points above the 1:1 line show areas of high leaching under present vegetation (fig 7b)

**Answer: Values <1 in Fig7a indicate regions where leaching of the potential natural vegetation is less than with land use. This corresponds to points Fig 7b above the 1:1 line. It is not useful to define the ratio the other way round, as leaching under natural vegetation can be very small but very large under land use (irrigation, fertilizer input). We have swapped the x and y axis in Fig7b. Now points are below the 1:1 in panel b for values<1 in panel a. This has been also applied for Fig8.**

Fig 9, where do the obs come from- especially for NPP and Veg C. Is each point supposed to represent an individual sites? This are from the same FluxNet sites as in the SI figures? Also, where do the observational error bars come from & how were they calculated? Finally, should correlation coefficients & significance be reported?

**Answer: The missing reference to the observational data has been added to the manuscript. The horizontal bars denote the minima and maxima of observed data belonging to the same LPJ grid cell.**

Section 4.1.3 Where are the LAI data shown that the addition of N biogeochemistry supposedly fixes? If this is the big advancement with the model presented here should these data also be shown? Is it just the addition of N biogeochemistry that's responsible for the proported improvements, or were other parametric or structural changes made?

**Answer: In previous versions without accounting for nitrogen dynamics, the intensity of crop production had been calibrated by scaling the maximum LAI value, the harvest index and the factor for scaling leaf-level photosynthesis to stand level (alpha_a) of each crop at the national scale as described in Fader et al. (2010). Without this calibration, yield levels are typically overestimated, especially in low-input regions (blue lines in Figures 10, S21-23). Significant parts of this calibration**

**are now no longer necessary, as N limitation already reduces simulated crop yields in low-input countries. However, the N limitation is not the only mechanism that leads to lower crop yields, so that a further calibration could be performed. However, we did not calibrate yields here to show that N limitation already leads to a better representation of low-input crop production. This is now better discussed in the text (now section 4.4.3).**

Fig. 10 A bunch of questions: What are the residuals and how are they calculated? What are units for the y-axis (what is the 'FM')? What are the little numbers in the top of each panel showing? Finally, it looks like global maize production increases over the period shown but the models are all flat. What's driving the increase in yields that the model is apparently missing? Is this true for other major crops?

**Answer: FM depicts "fresh matter" (as used for all data reported by the FAOstat) and tFM stands for ton fresh matter and is now explained in the figure caption. The residuals are the detrended observed and simulated values, by subtracting a moving average as described by Müller et al. (2017). The numbers in the panels show the correlation coefficients between the residuals of the FAO-stat data and the residuals of the simulation (dashed lines). This is now also explained in the text. Indeed, simulated crop yields show no trend, as all management assumptions (including fertilizer input) are static, whereas technological advances are driving the observed yield trend. This is now also discussed in the text.**

P 26 L 4, What improvements are necessary? What additional complexity may improve things further? What data are critical to getting terrestrial C-N dynamics less wrong? As presented these are kind of empty / throwaway statements. Can the be flushed out with some more detail, both in the main text and in this summary conclusion?

**Answer: Thanks for pointing this out. We now discuss further possible improvements in more detail, mainly incorporating aspects that have been implemented in parallel in the carbon-only version (LPJmL4.0, Schaphoff et al. 2017a,b), such as the improved tree phenology, but also more detail in the representation of agricultural management, such as tillage, or fertilizer type and timing.**

References in answers:

Ascott, MJ, DC Gooddy, L Wang, ME Stuart, MA Lewis, RS Ward, and AM Binley. 2017, Global patterns of nitrate storage in the vadose zone, Nat. Commun., 8, 1416, doi: 10.1038/s41467-017-01321-w.

Fader, M, S Rost, C Müller, A Bondeau, and D Gerten. 2010, Virtual water content of temperate cereals and maize: Present and potential future patterns, Journal of Hydrology, 384, 218-231, doi: 10.1016/j.jhydrol.2009.12.011.

Kabala, C, A Karczewska, B Galka, M Cuske, and J Sowinski. 2017, Seasonal dynamics of nitrate and ammonium ion concentrations in soil solutions collected using MacroRhizon suction cups, Environ Monit Assess, 189, 304, doi: 10.1007/s10661-017-6022-3.

Müller, C, J Elliott, J Chryssanthacopoulos, A Arneth, J Balkovic, P Ciais, D Deryng, C Folberth, M Glotter, S Hoek, T Iizumi, RC Izaurralde, C Jones, N Khabarov, P Lawrence, W Liu, S Olin, TAM Pugh, DK Ray, A Reddy, C Rosenzweig, AC Ruane, G Sakurai, E Schmid, R Skalsky, CX Song, X Wang, A de Wit, and H Yang. 2017, Global gridded crop model evaluation: benchmarking, skills, deficiencies and implications, Geosci. Model Dev., 10, 1403-1422, doi: 10.5194/gmd-10-1403-2017.

Schaphoff, S, M Forkel, C Müller, J Knauer, W von Bloh, D Gerten, J Jägermeyr, W Lucht, A Rammig, K Thonicke, and K Waha. 2017, LPJmL4 - a dynamic global vegetation model with managed land: Part II – Model evaluation, Geosci. Model Dev. Discuss., 2017, 1-41, doi: 10.5194/gmd-2017-146.

Schaphoff, S, W von Bloh, A Rammig, K Thonicke, H Biemans, M Forkel, D Gerten, J Heinke, J Jägermeyr, J Knauer, F Langerwisch, W Lucht, C Müller, S Rolinski, and K Waha. 2017, LPJmL4 – a dynamic global vegetation model with managed land: Part I – Model description, Geosci. Model Dev. Discuss., 2017, 1-59, doi: 10.5194/gmd-2017-145.

Walvoord, MA, FM Phillips, DA Stonestrom, RD Evans, PC Hartsough, BD Newman, and RG Striegl. 2003, A Reservoir of Nitrate Beneath Desert Soils, Science, 302, 1021.

References: Le Quéré C, et al. (2015) Global Carbon Budget 2015. Earth System

Science Data 7: 349-396. DOI:10.5194/essd-7-349-2015

Smith et al. (2014) Implications of incorporating N cycling and N limitations on primary

production in an individual-based dynamic vegetation model, Biogeosciences,

11, 2027-2054, doi:10.5194/bg-11-2027-2014.

Zaehle, et al. (2015). Nitrogen availability reduces CMIP5 projections of twenty-firstcentury

land carbon uptake. Journal of Climate 28.6: 2494-2511.

**Response to reviewer #2**

1. The results part before section 4.1 presents the comparison between outputs from global simulations with various model versions and setups with some interesting findings. However, the section is hard to follow without clear organization of the results, and some interesting findings and important characteristics are presented without proper investigations/explanations. I would suggest authors to make it a sub-section with clearer titles/points indicating the major content of each paragraph. A little deeper investigations or more detailed explanations on the differences between versions/setups would be preferable. In addition, model versions/setups should always be presented along with results (sometimes, it is missing).

**Answer: The results section has been restructured and expanded. We have now structured the results into 3 sections: Carbon pools and fluxes, Nitrogen pools and fluxes, Land use and nitrogen dynamics. A new figure plotting Taylotr diagrams of regional crop yields have been added to support our findings.**

2. Section 4.1.1 and 4.1.2 present the comparison of N and C pools/fluxes at both site-level and globally against literature values. Besides comparing values, some more explanations/discussions in the differences/discrepancies could be helpful for readers to understand the model performance.

**Answer: We have expanded the discussion of model results in various aspects e.g. by specifying better the effects of human land use, see also responses to concerns of reviewer #1.**

3. The authors present the improvement in simulating crop productivity by model developed in this study very briefly in Section 4.1.3, while it was shown as major results consisting almost half of the Abstract, which might make readers think the evaluation is only done for crop productivity. Even it is the major part of evaluation, I actually did not get the 'improvements' that authors claimed from the text. Though model results and its comparison with data were shown in Fig. 10 and lots of SI Figures, readers can only get little information on the model performance and the improvement due to implementation of N cycle (at least for me). With comparison on many sites, no statistically synthesis of the performance (no numbers), and no further investigations on 'how and why' were shown in the text. It is necessary/critical to show quantitatively how the improvement can be supported by the results. It is not the job of readers to check every figure, compare the lines, and draw the conclusion. In addition, there is no line for LPJmL3.5 in SI Fig. S5-S20, how the 'improvements' can be seen?

**Answer: We have now computed the average Willmott coefficients for the NEE and evaporation sites to get a quantitative estimate about the quality of our model. Also, we have extended the discussion on the crop model performance and added an additional figure to the main text and 3 more to the supplementary material. See also answer to the detailed point below.**

4. The authors tried to discuss the improvements and limitations of this study in 'Discussion and conclusions' section. But it is lack of clear structure. A synthesis of limitation might be helpful (e.g., listing limitations points by points at first).

**Answer: LPJmL5 lifts a major limitation of LPJmL3.5, which is the omission of nitrogen limitations in the simulation of plant growth and biogeochemical cycles. The implementation is based on well-established process descriptions reducing limitations in the form of assumptions to the former version..**

Minor comments (there might be some duplicate points as some listed above):

P18L1-3: Need to indicate the results come from which version/setup. The means of red/yellow lines were opposite in the text and the figure.

**Answer: The means of yellow/red lines is now in line with the figure.**

Fig. 4: It is necessary to show LPJmL35-PNV too. All lines in the figures of this manuscript should be drawn with thicker lines.

**Answer: LPJmL35-PNV has been added to Fig. 4 and line widths have been chosen thicker.**

P18L14: It seems not 'quite stable' and not 'slightly increase' in Fig. 5. And how the factor of 2 be derived?

**Answer: The terms "quite stable and "slightly "increase" have been replaced by quantitative numbers. In fact it is unclear how the factor 2 has been derived. The statement has been omitted from the manuscript.**

P18L22: Please give model setup.

**Answer: The model setup of the experiments is explained in the model setup section on page 16.**

P20L9-10:

What ratio? It is not clear. It would be necessary to explain why the ratios above 1 can occur. P21L9-13: I don't understand these 3 sentences. What's the meaning?

**Answer: Relative changes are calculated by dividing the values by their 1901-1910 average as now stated in the figure caption. If, e.g. GPP is higher than in the reference period, ratios >1 can occur.**

Figure 9: Please indicate the data sources for the observed values.

**Answer: A reference to the observed values has been added to the manuscript**

P23L5: It is necessary to show in SI figures the LPJmL3.5 output for a real comparison.

**Answer: We refer to the supplement of the evaluation paper of LPJmL4, where NEE and evaporation values are shown for all sites. A reference on page 23 has been added. The average of Willmott coefficient is shown for LPJML4 and LPJmL5 to allow a more quantitative comparison.**

P23L9: Please indicate what is 'satisfying', is there any indicators to draw this conclusion?

**Answer: We show now a quantitative estimate by using the Willmott coefficient of agreement.**

 P24L5: Please indicate where the climate conditions are not the only yield limiting factors.

**Answer: Crop yields can be limited by climatic conditions or nutrient supply, but also by outbreaks of pests and diseases, unstable social condition and others. We have added pest management as an example to the text.**

Sect. 4.1.3:

Please indicate how well the model performance. We can get nothing from the text.

**Answer: We have now extended the discussion and included an additional figure on crop performance in the main text and 3 in the supplementary document. As for all other crop models, crop model performance compared to FAO-stat data is very diverse and there is no single metric that would describe it well. The now added Taylor diagram may be the best single indicator that covers a broad range of aspects. The full crop model evaluation with the online tool at https://mygeohub.org/tools/ggcmevaluation is included as a zip file in the online material.**

[revised manuscript text omitted]
_{\text{sat}}) = \left( \frac{W_{\text{sat}} - b_{\text{nit}}}{a_{\text{nit}} - b_{\text{nit}}} \right)^{d_{\text{nit}} \cdot (b_{\text{nit}} - a_{\text{nit}})/(a_{\text{nit}} - c_{\text{nit}})} \cdot \left( \frac{W_{\text{sat}} - c_{\text{nit}}}{a_{\text{nit}} - c_{\text{nit}}} \right)^{d_{\text{nit}}}, \tag{38}$$

where $W_{\mathrm{sat}}$ is the water filled pore space of the soil, parameters $a_{\mathrm{nit}}$ to $d_{\mathrm{nit}}$ are given for sandy and medium soil (SI Table S1).

This soil pH function is based on Parton et al. (1996):

$$F(\mathrm{pH}) = 0.56 + \arctan(\pi \cdot 0.45 \cdot (-5 + \mathrm{pH}))/\pi \tag{39}$$

Soil pH values are taken from the WISE dataset (Batjes, 2000). Part of the N during the nitrification is lost to the atmosphere as nitrous oxide $N_2O$. Parton et al. (2001) assume that the $N_2O$ flux $F_{N_2O}$ (in gN m$^{-2}$ d$^{-1}$) is proportional to the nitrification rate with

$$F_{N_2O} = K_2 \cdot F_{NO_3^-}, \tag{40}$$

where $K_2$ is fraction of nitrified N lost as $N_2O$ flux ($K_2 = 0.02$). Finally, soil $NO_3^-$ and $NH_4^+$ are updated accordingly:

$$\begin{aligned}
NO_{3,\mathrm{soil},l,t+1}^- &= NO_{3,\mathrm{soil},l,t}^- + (1 - K_2) \cdot F_{NO_3^-} & (41)\\
NH_{4,\mathrm{soil},l,t+1}^+ &= NH_{4,\mathrm{soil},l,t}^+ - F_{NO_3^-} & (42)
\end{aligned}$$

**2.6.4 Denitrification**

The reduction of $NO_3^-$ to $NO_2$ and $N_2$ is determined for each soil layer using the implementation in SWIM (Krysanova and Wechsung, 2000).

$$D_{NO_3^-} = F_2(W_{\mathrm{sat}}) \cdot F_2(T_{\mathrm{soil}}, C_{\mathrm{org}}) \cdot NO_{3,\mathrm{soil}}^-, \tag{43}$$

where $F_2(W_{\mathrm{sat}})$ is the water response function and $F_2(T, C)$ the soil temperature and carbon reaction function. The water response function depends on the water filled pore space $W_{\mathrm{sat}}$ in the following way:

$$F_2(W_{\mathrm{sat}}) = 6.664096 \times 10^{-10} \cdot \exp(21.12912 \cdot W_{\mathrm{sat}}) \tag{44}$$

The water response function shows a qualitatively similar behavior to Eq. 151 from SWIM while ensuring continuity (see SI Fig. S4). Parameters are fitted and adjusted so that for full soil water saturation, the value is not greater than 1. The soil temperature and carbon reaction function is parameterized according to:

$$F_2(T_{\mathrm{soil}}, C_{\mathrm{org}}) = 1 - \exp(-\mathrm{CDN} \cdot F_2(T_{\mathrm{soil}}) \cdot C_{\mathrm{org}}), \tag{45}$$

where $\mathrm{CDN} = 1.4$ is the shape coefficient (Arnold et al., 2012), $C_{\mathrm{org}}$ is the sum of the fast and slow C pools and $F_2(T_{\mathrm{soil}})$ is the soil temperature reaction function. $F_2(T_{\mathrm{soil}})$ is replaced by Equation C5 from Smith et al. (2014) which is only valid for positive $T_{\mathrm{soil}}$. The original function from SWAT approaches 1 for high temperatures whereas the function from Smith declines which seems more sensible. Equation C5 of Smith et al. (2014) is taken from Comins and McMurtrie (1993).

$$F_2(T_{\mathrm{soil}}) = \begin{cases} 0.0326 & \text{for } T_{\mathrm{soil}} \leq 0^\circ\mathrm{C} \\ 0.0326 + 0.00351 \cdot T_{\mathrm{soil}}^{1.652} - (T_{\mathrm{soil}}/41.748)^{7.19} & \text{for } 0^\circ\mathrm{C} < T_{\mathrm{soil}} < 45.9^\circ\mathrm{C} \\ 0 & \text{for } T_{\mathrm{soil}} \geq 45.9^\circ\mathrm{C} \end{cases} \tag{46}$$

Bessou et al. (2010) assume that the N$_2$O flux from NO$_3^-$, $F_{\mathrm{N_2O}}$ (in gN m$^{-2}$ d$^{-1}$), is proportional to the denitrification rate $D_{\mathrm{NO_3^-}}$ with

$$F_{\mathrm{N_2O}} = r_{\mathrm{mx}} \cdot D_{\mathrm{NO_3^-}}, \tag{47}$$

where $r_{\mathrm{mx}} = 0.11$ is the fraction of denitrified N lost as N$_2$O flux. The N$_2$ flux $F_{\mathrm{N_2}}$ is then derived by

5  $$F_{\mathrm{N_2}} = (1 - r_{\mathrm{mx}}) \cdot D_{\mathrm{NO_3^-}}, \tag{48}$$

The soil NO$_3^-$ pools have to be reduced by the denitrification flux:

$$\mathrm{NO_{3,soil,}}_{l,t+1}^- = \mathrm{NO_{3,soil,}}_{l,t}^- - D_{\mathrm{NO_3^-}} \tag{49}$$

**2.6.5 Nitrogen leaching and movement**

Nitrate movement with water fluxes is simulated as in SWAT (Neitsch et al., 2002, 2005). Nitrate is assumed to be fully
10  dissolved in water and moves with surface runoff, lateral runoff and percolation water. To compute the amount of nitrate transported with the water from a soil layer, we first calculate the concentration of nitrate in the mobile water. This concentration is then multiplied by the volume of surface runoff, lateral runoff or percolation water between soil layers or into the aquifer, respectively. The amount of nitrate leached depends on the climatic and soil conditions and on the type and intensity of soil management (e.g. plant cover, soil treatment, fertilization).

15  The concentration of nitrate in the mobile water $\mathrm{conc}_{\mathrm{NO_3^-,mobile,}l}$ in layer $l$ (kgN m$^{-3}$) is:

$$\mathrm{conc}_{\mathrm{NO_3^-,mobile,}l} = \frac{\mathrm{NO_{3,soil,}}_l^- \cdot \left(1 - \exp\left(\frac{-w_{\mathrm{mobile,}l}}{(1-\theta)\cdot\mathrm{SAT}_l}\right)\right)}{w_{\mathrm{mobile,}l}}, \tag{50}$$

where $\mathrm{NO_{3,soil,}}_l^-$ is the content of nitrate in layer $l$ (gN m$^{-2}$), $w_{\mathrm{mobile}}$ is the amount of mobile water in the layer (mm), $\theta = 0.4$ is the fraction of porosity from which anions are excluded (0.5 in Neitsch et al., 2002), and $\mathrm{SAT}_l$ is the saturated water content of the soil layer (mm).

20  The mobile water $w_{\mathrm{mobile,}l}$ in the layer $l$ is the amount of water lost by surface runoff, lateral flow and percolation:

$$w_{\mathrm{mobile,}l} = \begin{cases} Q_{\mathrm{surf}} + Q_{\mathrm{lat,}l=1} + w_{\mathrm{perc,}l=1} & \text{for } l = 1 \\ Q_{\mathrm{lat,}l} + w_{\mathrm{
[revised manuscript text omitted]

---

## Author Response (AR2)

**Response to reviewer**

I appreciate the revisions by vonBloh and co-authors. Remaining concerns are largely technical in nature, but it seems appropriate in the introduction and methods to contextualize the approach for representing N limitation in a global model compared to a growing list of ESMs that tackle the same challenge using a variety of approaches.

Major Concerns:

I'm still uncertain how N limitation actually occurs in the model, this refers to Section 2.4 of the text and Fig 2. The flow chart for Fig 2 seems to describe how GPP is calculated, but the text for the section is all about NPP, please clarify. The description of calculating a water stress, but N unlimited Vmax and photosynthesis rate and subsequently calculating the N limited Vmax seems very similar to CLM4cn (Thornton et al. 2007, referenced in the text). In CLM, however, this resulted in a decoupling of leaf-level water and C exchange. Is the same approach being taken here, or does leaf gas exchange based on the N limited Vmax calculated after N limitation is accounted for?

**Answer: We agree with the concerns of the reviewer and have changed our calculation scheme: After determining the nitrogen-limited Vmax we recalculate the photosynthesis rate and the corresponding transpiration rates. The change lead to an improvement of the NEP and evaporation fluxes.**

Minor and Editorial concerns

Page 2 line 7-9 As with the major concern, above, this sentence still seems awkward. This strikes me as an opportunity to clarify the approach taken with LPJmL5. For example, if the current 'implementation is based on previous model implementations', what are they? How is the LPJmL5 approach similar and how is it different. There are several ways to simulate N limitation in global scale models (referenced on the previous page), but the sentence here only casually describes what's being done without giving much information to the reader (e.g., 'N limitation occurs, not by XXX…', or something that briefly and accurately describe how plant and soil N availability effects the terrestrial C cycle in the model). A few sentences providing a broad overview the N approach taken here will help clarify the contributions made here.

**Answer: We have listed the new functionality (soil nitrogen dynamics, plant uptake, nitrogen allocation, response of photosynthesis, transpiration and maintenance respiration) in the sentence before the one highlighted here. We have now added references to the main model**

**implementations from which we have drawn our approaches here and now point out that we discuss the implementation and the corresponding references in full detail in the following sections.**

The list of references in Table 2 is pretty overwhelming, and not terribly helpful. If the purpose of these references is for data reproducibility, it's not at all clear what data were used for particular parameter values or PFTs. Are these the citation needed to satisfy the TRY data use requirements, or were they collected in addition to the (Kattge et al., 2011) reference?

**Answer: The policy of the TRY database requires the inclusion of all references where the data used in the paper comes from.**

Table S1 should have a heading above, not below the table.

**Answer: This has now been corrected in the supplement.**

Section 2.6 & Fig. 3. I should have noticed this earlier, but in I'm not clear how the decomposition of SOM directly liberates $NO_3$ during (Knit on Fig. 3, 'Fraction of mineralized N nitrified to $NO-3$, which = 0.2 in Table S1). It seems this value should be 0 by definition, and not shown on the figure. For example, in the Schimel and Bennett (2004) model of inorganic N transformations, or Davidson's 1991 Leaky pipe suggest that $NO_3$ formation only occurs through nitrification. Is the direct flux from SOM to $NO_3$ ecologically justified, or is commonly it represented in other models? Looking at Parton et al. 2001, it does seem like 20% of mineralized N is sent to the $NO_3$ pool in DAYCENT, but is this just a mathematical artifact of the sequential solver used in the models that allows $NO_3$ losses to occur? It seems like an odd 'feature' of the model to perpetuate?

**Answer: The fraction of mineralized N nitrified to $NO-3$ has now been set to zero following (Schimel and Bennett, 2004).**

Section 2.6.2 Gerber didn't work on CLM, to my knowledge.

**Answer: The reviewer is right; we have replaced the CLM model by the LM3V model used by Gerber.**

Section 2.6.4 What is SWAT? The acronym should be defined in the text.

**Answer: The acronym SWAT (soil and water assessment tools) has now been explained in the paper.**

Section 2.6.5 If NO3 leaching is calculated sequentially (after denitrification) why does the abstract and main text focus on the hydrologic losses of inorganic N and not the gaseous losses simulated by the model? (See also table 4).

**Answer: We have adjusted the sequence of processes listed in section 2.6 to the sequence in which these are computed in the model. As all of these are relatively small daily rates, the sequence of implementation does not affect results too much as the first function called responds to the concentrations after the last function called from the previous time step (day before) and so on. We discuss leaching in the abstract, as this is the largest nitrogen loss flux and also the one that responds most strongly to the inclusion of human land use.**

The colors, symbols and acronyms included in Fig 10 are so complicated as to preclude any meaningful insight from the display item. The yellow star we're supposed to compare to red circle cross is nearly impossible to find. More, the caption doesn't help explain the complexity in a way to aid readers in interpretation of the figure, thus as presented I'd recommend removing it from the main text. Alternatively, the display item can stand, but more information beyond the Müller citation is needed for readers to understand the information communicated in the context of the work presented.

**Answer: Following the suggestion of the reviewer we have put Fig. 10 in the supplement.**

**We have made the source code of our model public available under http://doi.org/10.5880/pik.2018.011**

[revised manuscript text omitted]
_{\text{NC}}(\text{NC}_{\text{plant}}) = \frac{\text{NC}_{\text{leaf,high}} - \text{NC}_{\text{plant}}}{\text{NC}_{\text{leaf,high}} - \text{NC}_{\text{leaf,low}}}, \tag{10}$$

where $\text{NC}_{\text{leaf,low}}$ and $\text{NC}_{\text{leaf,high}}$ are the lower and upper limits of N:C ratios and $\text{NC}_{\text{plant}}$ is the actual plant N:C ratio. The lower and upper limits $\text{NC}_{\text{leaf,low}}$ and $\text{NC}_{\text{leaf,high}}$ are derived from the TRY database (Kattge et al., 2011). Their reciprocal

C:N values for each PFT are shown in Table 2. The actual plant N:C ratio is calculated according to

$$NC_{plant} = \frac{N_{leaf} + N_{root}}{C_{leaf} + C_{root}} \tag{11}$$

The temperature function $f_T$ for N uptake is given by Thornley (1991):

$$f_T(T_{soil,l}) = (T_{soil,l} - T_0) \cdot (2 \cdot T_m - T_0 - T_{soil,l})/(T_r - T_0)/(2 \cdot T_m - T_0 - T_r), \tag{12}$$

where $T_0 < T_r < 2 \cdot T_m - T_0$. For the chosen $T_m = 15°C$, $T_r = 15°C$ and $T_0 = -25°C$, the maximum of 1 is reached at $15°$ and the function is positive above -25°C.

The root distribution  rootdist$_l$ can be calculated from the proportion of roots from surface to soil depth $z$,  rootdist$_z$, as in Jackson RB et al. (1996):

$$rootdist_z = \frac{\int_0^z (\beta_{root})^{z'} dz'}{\int_0^{z_{bottom}} (\beta_{root})^{z'} dz'} = \frac{1 - (\beta_{root})^z}{1 - (\beta_{root})^{z_{bottom}}}, \tag{13}$$

where $\beta_{root}$ is a PFT-specific parameter (for parameter values see Table 2).  rootdist$_l$ is then given by the difference  rootdist$_{z(l)}$ − rootdist$_{z(l-1)}$. If the soil depth of the layer $l$ is greater than the thawing depth then rootdist$_l$ is reduced accordingly. The non-zero rootdist$_l$ are rescaled so that their sum is normalized to one, accounting for the modified root distribution under freezing conditions. Soil $NH_4^+$ and soil $NH_3^-$ pools are reduced accordingly every simulation day $t$:

$$NO_{3,soil,l,t+1}^- = NO_{3,soil,l,t}^- \cdot \left(1 - rootdist_l \cdot \frac{N_{uptake}}{\sum_{l=1}^{n_{soillayer}} N_{avail,l}}\right) \tag{14}$$

$$NH_{4,soil,l,t+1}^+ = NH_{4,soil_l,t}^+ \cdot \left(1 - rootdist_l \cdot \frac{N_{uptake}}{\sum_{l=1}^{n_{soillayer}} N_{avail,l}}\right) \tag{15}$$

**2.3 Determination of the N limitation scalar**

For trees, grass and crops, the N limitation scalar $v_{scal}$ is calculated  as the ratio of N demand $N_{uptake,opt}$  to actual N uptake:

$$v_{scal} = \min(N_{uptake}/N_{uptake,opt}, 1) \tag{16}$$

The scalar $v_{scal}$ is used to account for N limitation in the allocation of N to different plant organs (section 2.5) and is computed as the growing season mean, which is re-initialized to zero every year for natural vegetation and at sowing for crops.

**2.4  Photosynthesis, gross and net primary production under N limitation**

To calculate the limitation by N availability, N stress is calculated after determining water stress on photosynthesis. If N demand from the water-limited photosynthesis rate cannot be fulfilled by N uptake, carboxylation capacity $V_{max}$ has to be reduced. The reduced $V_{max}$ is determined by solving Eq. (1) for $V_{max}$. Water demand is then recalculated using the reduced

**Table 2.** PFT-specific $\beta_{\text{root}}$ based on Schaphoff et al. (2018b) and minimum and maximum leaf C:N ratios, based on the TRY data base (Kattge et al., 2011) with data from Kurokawa and Nakashizuka (2008); Garnier et al. (2007); Penuelas et al. (2010a); Fyllas et al. (2009); Loveys et al. (2003); Han et al. (2005); Ordonez et al. (2010); Atkin et al. (1999); White et al. (2000); Xu and Baldocchi (2003); Freschet et al. (2010a, b); Laughlin et al. (2010); Niinemets (2001, 1999); Willis et al. (2010); Baker et al. (2009); Patino et al. (2009); Pakeman et al. (2009, 2008); Fortunel et al. (2009); Penuelas et al. (2010b); Cornelissen et al. (1996, 1997, 2004); Quested et al. (2003); Sardans et al. (2008b, a); Ogaya and Penuelas (2003, 2006, 2007, 2008). The C:N ratios for C3 and C4 grasses and crops are based on White et al. (2000).

| Functional type | $CN_{\text{leaf,low}}$ | $CN_{\text{leaf,high}}$ | $\beta_{\text{root}}$ |
|---|---|---|---|
| Tropical broadleaved evergreen tree | 15.6 | 46.2 | 0.962 |
| Tropical broadleaved raingreen tree | 15.4 | 34.6 | 0.961 |
| Temperate needleleaved evergreen tree | 31.8 | 63.8 | 0.976 |
| Temperate broadleaved evergreen tree | 15.6 | 46.2 | 0.964 |
| Temperate broadleaved summergreen tree | 15.4 | 34.6 | 0.966 |
| Boreal needleleaved evergreen tree | 31.8 | 63.8 | 0.943 |
| Boreal broadleaved summergreen tree | 15.4 | 34.6 | 0.943 |
| Boreal needleleaved summergreen tree | 18.4 | 36.9 | 0.943 |
| C3 perennial grass | 10.5 | 37.9 | 0.972 |
| C4 perennial grass | 17.4 | 66.9 | 0.943 |
| Bioenergy tropical tree | 15.6 | 46.2 | 0.976 |
| Bioenergy temperate tree | 15.4 | 34.6 | 0.976 |
| Bioenergy C4 grass | 17.4 | 66.9 | 0.976 |
| Crops | 14.3 | 58.8 | 0.972 |

$V_{\text{max}}$. From this reduced $V_{\text{max}}$, the actual photosynthesis rate and canopy conductance can be calculated (Fig. 2). For the determination of the canopy conductance we assume higher PFT-specific minimum canopy conductances $g_{\text{min}}$ (see SI Table S1) than Schaphoff et al. (2018b), which are in the range of values reported by Barnard and Bauerle (2013). Furthermore we have adjusted some additional parameters (SI Table S1, S2) to meet global and local evapotranspiration fluxes under nitrogen limitation effects on transpiration. The gross primary production (GPP) derived from the actual photosynthesis rate is reduced by leaf, root and sapwood (for tree PFTs) respiration $R_{\text{leaf}}$, $R_{\text{root}}$, and $R_{\text{sapwood}}$ in order to get the net primary production (NPP). Respiration rates of roots and sapwood is are assumed to be linearly dependent on the N:C ratio of the corresponding pool, whereas the respiration rate of leaves ($R_{\text{leaf}}$) is a fraction (1.5% for C3 plants, 3.5% for C4 plants) of $V_{\text{max}}$ (Sitch et al., 2003):

$$R_{\text{root}} = k_{\text{resp}}(T_{\text{soil}}) \cdot N_{\text{root}} \tag{17}$$

$$R_{\text{sapwood}} = k_{\text{resp}}(T_{\text{air}}) \cdot N_{\text{sapwood}}, \tag{18}$$

[Figure]

**Figure 2.** Calculation of N stress of plants.

where $k_{\mathrm{resp}}(T)$ is a temperature dependent respiration rate (gC gN$^{-1}$ d$^{-1}$) (as in Sitch et al., 2003). Therefore higher N:C ratios lead to a reduction in net primary production (NPP), which is computed as:

$$\mathrm{NPP} = \mathrm{GPP} - R_{\mathrm{growth}} - R_{\mathrm{leaf}} - R_{\mathrm{root}} - R_{\mathrm{sapwood}}, \tag{19}$$

[revised manuscript text omitted]

where the annual shift rates $N^{s,f}_{\text{shift},l}$ describe the organic matter input from the different PFTs into the respective layer due to cryoturbation and bioturbation (Schaphoff et al., 2013).

Net mineralized material $N_{\text{miner,litter},l}$ is

$$N_{\text{miner,litter},l} = A_f \cdot N_{\text{decom}} \cdot (F_f \cdot N^f_{\text{shift},l} + (1 - F_f) \cdot N^s_{\text{shift},l}), \tag{27}$$

which adds  N to an intermediate N mineralization pool

$$N_{\text{miner},l} = N_{\text{miner,soil},l} + N_{\text{miner,litter},l} \tag{28}$$

 In contrast to Parton et al. (2001) where 20% of this pool is  directly nitrified to $NO_3^-$  Parton et al. (2001)) , we follow Schimel and Bennett (2004) and transfer all mineralized N to the $NH_4^+$

$$NH^+_{4,\text{soil},l,t+1} \equiv NH^+_{4,\text{soil},l,t} + (1 - K_{\text{nit}}) \cdot N_{\text{miner},l}$$

$$NO^-_{3,\text{soil},l,t+1} \equiv NO^-_{3,\text{soil},l,t} + K_{\text{nit}} \cdot N_{\text{miner},l}$$

pool:

$$NH^+_{4,\text{soil},l,t+1} = NH^+_{4,\text{soil},l,t} + N_{\text{miner},l} \tag{29}$$

**2.6.2 Nitrogen immobilization**

Immobilization, i.e. the transformation of mineral N to organic N in soils, is determined per soil layer directly after soil and litter mineralization, following the  LM3V land model described by Gerber et al. (2010). If available mineral soil N is constraining immobilization, mineral N is first immobilized into the fast soil pool and then into the slow soil pool.

5   The immobilized N $N_{\mathrm{immo},l}$ is calculated according to

$$N_{\mathrm{immo},l} = F_f \cdot (1 - A_f) \cdot (C_{\mathrm{decom}}/\mathrm{CN}_{\mathrm{soil}} - N_{\mathrm{decom}}) \cdot N^f_{\mathrm{shift},l} \cdot \frac{N_{\mathrm{sum},l}/d_{\mathrm{soil},l}}{k_N + N_{\mathrm{sum},l}/d_{\mathrm{soil},l}}, \tag{30}$$

where $\mathrm{CN}_{\mathrm{soil}}$ is the desired soil C:N ratio of 15 (dimensionless) for all soil types, $d_{\mathrm{soil},l}$ is the soil depth of layer $l$ in m, $k_N = 5 \times 10^{-3}$ (gN m$^{-3}$) is the half saturation concentration for immobilization in soils (Gerber et al., 2010), and $N^f_{\mathrm{shift},l}$ is the parameter that determines the distribution of the humified organic matter in the topsoil to the different soil layers $l$

10  (Schaphoff et al., 2013). The available mineral N in the soil layer $l$ ($N_{\mathrm{sum},l}$ in gN m$^{-2}$) is the sum of $\mathrm{NH_4^+}$ and $\mathrm{NO_3^-}$:

$$N_{\mathrm{sum},l} = \mathrm{NH^+_{4,soil},l} + \mathrm{NO^-_{3,soil},l} \tag{31}$$

The immobilized N ($N_{\mathrm{immo},l}$) is added to the fast soil N pool of layer $l$ and subtracted from the $\mathrm{NH_4^+}$ and $\mathrm{NO_3^-}$ pools:

$$
\begin{aligned}
P^f_{\mathrm{soil},l,t+1} &= P^f_{\mathrm{soil},l,t} + \min(N_{\mathrm{immo},l}, N_{\mathrm{sum},l}) & (32)\\
\mathrm{NH^+_{4,soil},l,t+1} &= \mathrm{NH^+_{4,soil},l,t} - \mathrm{NH^+_{4,soil},l,t} \cdot \min(N_{\mathrm{immo},l}/N_{\mathrm{sum},l}, 1) & (33)\\
\mathrm{NO^-_{3,soil},l,t+1} &= \mathrm{NO^-_{3,soil},l,t} - \mathrm{NO^-_{3,soil},l,t} \cdot \min(N_{\mathrm{immo},l}/N_{\mathrm{sum},l}, 1) & (34)
\end{aligned}
$$

The immobilization into the slow soil N pool ($P^s_{\mathrm{soil},l,t+1}$) is computed accordingly as in Eq. (30) but with $(1 - F_f)$ instead of $F_f$.

**2.6.3 Nitrification**

Nitrogen fluxes from nitrification in the soil are modeled modified after Parton et al. (2001) with the schematic representation
20  of a series of pipes for the main flow from $\mathrm{NH_4^+}$ over $\mathrm{NO_3^-}$ to $\mathrm{N_2}$ from which $\mathrm{N_2O}$ leaks in between. As suggested by Parton et al. (2001, equation 2), nitrification is computed as a fixed fraction of the mineralization flux (see 2.6.1) as well as an explicit transformation flux $F_{\mathrm{NO_3^-}}$ from ammonium to nitrate in gN m$^{-2}$ d$^{-1}$, which is described here.

$$F_{\mathrm{NO_3^-}} = K_{\max} \cdot F_1(T_{\underset{\sim}{\mathrm{soil}}\mathrm{soil},l}) \cdot F_1(W_{\underset{\sim}{\mathrm{sat}}\mathrm{sat},l}) \cdot F(\mathrm{pH}) \cdot \mathrm{NH^+}_{4,\underset{\sim}{\mathrm{soil}}4,\mathrm{soil},l}, \tag{35}$$

where  $\mathrm{NH^+_{4,soil,l}}$ is the model-derived soil ammonium concentration (gN m$^{-2}$), $K_{\max}$ is the maximum nitrification
25  rate of $\mathrm{NH_4^+}$ ($K_{\max} = 0.1$ d$^{-1}$),  $F_1(T_{\mathrm{soil},l})$ is the limiting function for temperature and  $F_1(W_{\mathrm{sat},l})$ the corresponding limiting function for water saturation  $W_{\mathrm{sat},l}$. Parton et al. (2001) show nitrification rates after data of Malhi and McGill (1982) in Table 3 without a formula. Using these data from three different sites in the US, Canada and Australia, we fitted a bell shaped function for the temperature dependence:

$$F_1(T_{\underset{\sim}{\mathrm{soil}}\mathrm{soil},l}) = \exp(-(T_{\underset{\sim}{\mathrm{soil}}\mathrm{soil},l} - a)^2/(2 \cdot b^2)), \tag{36}$$

where $a = 18.79°C$ and $b = 5.26$ give the best fist to the data (see SI Fig. S3). The function is applicable also for negative values.

The soil water response function $F_1(W_{sat})$ is parameterized according to Doran et al. (1988) as described in Parton et al. (1996):

$$F_1(W_{\underline{sat}sat,l}) = \left( \frac{W_{sat} - b_{nit}}{a_{nit} - b_{nit}} \frac{W_{sat,l} - b_{nit}}{a_{nit} - b_{nit}} \right)^{d_{nit} \cdot (b_{nit} - a_{nit})/(a_{nit} - c_{nit})} \cdot \left( \frac{W_{sat} - c_{nit}}{a_{nit} - c_{nit}} \frac{W_{sat,l} - c_{nit}}{a_{nit} - c_{nit}} \right)^{d_{nit}}, \tag{37}$$

where $\underline{W_{sat}}W_{sat,l}$ is the water filled pore space of  soil layer $l$, parameters $a_{nit}$ to $d_{nit}$ are given for sandy and medium soil (SI Table S2).

This soil pH function is based on Parton et al. (1996):

$$F(pH) = 0.56 + \arctan(\pi \cdot 0.45 \cdot (-5 + pH))/\pi \tag{38}$$

Soil pH values are taken from the WISE dataset (Batjes, 2000). Part of the N during the nitrification is lost to the atmosphere as nitrous oxide $N_2O$. Parton et al. (2001) assume that the $N_2O$ flux $F_{N_2O}$ (in gN m$^{-2}$ d$^{-1}$) is proportional to the nitrification rate with

$$F_{N_2O} = K_2 \cdot F_{NO_3^-}, \tag{39}$$

where $K_2$ is fraction of nitrified N lost as $N_2O$ flux ($K_2 = 0.02$). Finally, soil $NO_3^-$ and $NH_4^+$ are updated accordingly:

$$NO_{3,soil,l,t+1}^- = NO_{3,soil,l,t}^- + (1 - K_2) \cdot F_{NO_3^-} \tag{40}$$

$$NH_{4,soil,l,t+1}^+ = NH_{4,soil,l,t}^+ - F_{NO_3^-} \tag{41}$$

**2.6.4 Denitrification**

The reduction of $NO_3^-$ to $NO_2$ and $N_2$ is determined for each soil layer using the implementation in SWIM (Krysanova and Wechsung, 2000).

$$D_{NO_3^-} = F_2(W_{\underline{sat}sat,l}) \cdot F_2(T_{\underline{soil}soil,l}, C_{\underline{org}org,l}) \cdot NO_{3,\underline{soil}3,soil,l}^-, \tag{42}$$

where $\underline{F_2(W_{sat})}F_2(W_{sat,l})$ is the water response function and $F_2(T, C)$ the soil temperature and carbon reaction function. The water response function depends on the water filled pore space $\underline{W_{sat}}W_{sat,l}$ in the following way:

$$F_2(W_{\underline{sat}sat,l}) = 6.664096 \times 10^{-10} \cdot \exp(21.12912 \cdot W_{\underline{sat}sat,l}) \tag{43}$$

The water response function shows a qualitatively similar behavior to Eq. 151 from SWIM while ensuring continuity (see SI Fig. S4). Parameters are fitted and adjusted so that for full soil water saturation, the value is not greater than 1. The soil temperature and carbon reaction function is parameterized according to:

$$F_2(T_{\underline{soil}soil,l}, C_{\underline{org}org,l}) = 1 - \exp(-CDN \cdot F_2(T_{\underline{soil}soil,l}) \cdot C_{\underline{org}org,l}), \tag{44}$$

where CDN $= 1.4$ is the shape coefficient (Arnold et al., 2012), $\color{red}{C_\text{org}}\color{black}C_{\text{org},l}$ is the sum of the fast and slow C pools and $\color{red}{F_2(T_\text{soil})}\color{black}F_2(T_{\text{soil},l})$ is the soil temperature reaction function. $\color{red}{F_2(T_\text{soil})}\color{black}F_2(T_{\text{soil},l})$ is replaced by Equation C5 from Smith et al. (2014) which is only valid for positive $\color{red}{T_\text{soil}}\color{black}T_{\text{soil},l}$. The original function from $\color{red}{\text{SWAT}}$ the soil and water assessment tool (SWAT) approaches 1 for high temperatures whereas the function from Smith declines which seems more sensible. Equation C5 of Smith et al. (2014) is taken from Comins and McMurtrie (1993).

$$F_2(T_{\underset{\sim}{\text{soil}}\text{soil},l}) = \begin{cases} 0.0326 & \text{for } T_{\text{soil},l} \leq 0°\text{C} \\ 0.0326 + 0.00351 \cdot T_{\text{soil},l}^{1.652} - (T_{\text{soil},l}/41.748)^{7.19} & \text{for } 0°\text{C} < T_{\text{soil},l} < 45.9°\text{C} \\ 0 & \text{for } T_{\text{soil},l} \
[revised manuscript text omitted]

**S1    Supplementary information to the nitrogen implementation of the LPJmL5 model**

The supplement contains  tables of parameters used in the model (Table  S1 and S2) and graphical representations of leaf C:N ratios for different exponential factors (Fig. S1), daily gross photosynthesis rate as a function of light- and Rubisco limited photosynthesis rate (Fig. S2), temperature response function $F_1(T)$ (Fig. S3), and water response function $F_2(W)$ (Fig. S4).
Furthermore comparisons of net ecosystem exchange rates and evaporation fluxes with EDDY flux tower measurements (**?**) for a variety of sites are shown (Figs. S5-S11 and Figs. S12-S20, respectively). Taylor diagrams of the spatial patterns (national mean yields) of wheat, maize, rice and soybean are plotted in Figs. S21-S24. Wheat, rice and soybean simulations for the 10 top-producing countries for the carbon-only LPJmL3.5 version, the version with N limitation and with unlimited N supply are shown in Figs. S25-S27.

[Figure]

**Figure S1.** Leaf C:N ratio of the canopy as a function of LAI for different pre-factors of the exponential term in Eq. (2).

**Table S1.** List of PFT-specific parameters (maximum N uptake rate $N_{\mathrm{up,root}}$, increase in N demand $k_{\mathrm{store}}$, N recover fraction at turnover $k_{\mathrm{turn}}$, minimum canopy conductance $g_{\mathrm{min}}$, maximum water transport capacity $E_{\mathrm{max}}$) used in the LPJmL5 model.

| PFT | $N_{\mathrm{up,root}}$ (gN kgC$^{-1}$) | $k_{\mathrm{store}}$ | $k_{\mathrm{turn}}$ (%) | $g_{\mathrm{min}}$ (mm s$^{-1}$) | $E_{\mathrm{max}}$ (mm day$^{-1}$) |
|---|---|---|---|---|---|
| Tropical broadleaved evergreen tree | 2.8 | 1.15 | 80 | 1.6 | 10 |
| Tropical broadleaved raingreen tree | 2.8 | 1.15 | 30 | 1.8 | 10 |
| Temperate needleleaved evergreen tree | 2.8 | 1.15 | 80 | 1.0 | 7 |
| Temperate broadleaved evergreen tree | 2.8 | 1.15 | 80 | 1.5 | 7 |
| Temperate broadleaved summergreen tree | 2.8 | 1.15 | 30 | 1.0 | 7 |
| Boreal needleleaved evergreen tree | 2.8 | 1.15 | 80 | 0.8 | 7 |
| Boreal broadleaved summergreen tree | 2.8 | 1.15 | 30 | 0.8 | 7 |
| Boreal needleleaved summergreen tree | 2.8 | 1.15 | 30 | 0.3 | 7 |
| C3 perennial grass | 5.1 | 1.3 | 30 | 0.8 | 7 |
| C4 perennial grass | 5.1 | 1.3 | 30 | 1.5 | 10 |
| All crops | 5.1 | 1.3 | 30 | 0.8 | 8 |

[Figure]

**Figure S2.** Daily gross photosynthesis rate $A_{\mathrm{gd}}$ as a function of Rubisco-limited photosynthesis rate $J_C$ for fixed light-limited rate $J_E$ and daylength set to 1. The black solid curve is for shape parameter $\theta = 1$ ($A_{\mathrm{gd}} = \min(J_E, J_C)/\mathrm{daylength}$), the blue curve for $\theta = 0.9$ (LPJmL5), the red curve for $\theta = 0.7$ (LPJmL3.5) and the black dashed curve for $\theta \to 0$ ($A_{\mathrm{gd}} = J_E \cdot J_C/((J_E + J_C) \cdot \mathrm{daylength})$).

**Table S2.** List of parameters used in the LPJmL5 model.

| Parameter | De |
|---|---|
| $\alpha_a$ | Fr |
| $\alpha_m$ | M |
| $g_m$ | Co |
| $\theta$ | Sh |
|  $K_{\text{N,min}}$ | M |
| $k_{\text{N,min}}$ | Ba |
|   $f_{\text{heartwood}}$ | Fr |
| $K_M$ | M |
| $K_{\text{max}}$ | M |
| $a$ | Pa |
| $b$ | Pa |
| $a_{\text{nit}}$ | Pa |
| $b_{\text{nit}}$ | Pa |
| $c_{\text{nit}}$ | Pa |
| $d_{\text{nit}}$ | Pa |
| CDN | Sh |
| $A_f$ | Fr |
| $F_f$ | Fr |
| $\beta_{\text{NO}_3^-}$ | N |
| $\text{CN}_{\text{soil}}$ | D |
| $k^f_{\text{soil10}}$ | D |
| $k^s_{\text{soil10}}$ | D |
|  $r_{\text{mx}}$ | Fr |
| $\theta$ | Fr |
| $K_2$ | Fr |
| $K_N$ | M |
| $T_0$ | Pa |
| $T_m$ | Pa |
| $T_r$ | Pa |
| $q_{\text{ash}}$ | Fr |

[Figure]

**Figure S3.** Temperature response data for site in the US (filled squares), Canada (filled circles) and Australia (filled triangles) and fitted function (solid line) used in Eq. (36).

[Figure]

**Figure S4.** Water response functions $F_2(W_{sat})$ parameterized according to **?** (dashed line) and according to Eq. (43) in the main text (solid line).

[Figure]

**Figure S5.** Comparison of net ecosystem exchange rates (NEE, in gC m$^{-2}$ d$^{-1}$) simulated with eddy flux tower rates measured, $W$ denotes the Willmott coefficient of agreement.

[Figure]

**Figure S6.** Comparison of net ecosystem exchange rates (NEE, in gC m$^{-2}$ d$^{-1}$) simulated with eddy flux tower rates measured, $W$ denotes the Willmott coefficient of agreement.

[Figure]

**Figure S7.** Comparison of net ecosystem exchange rates (NEE, in gC m$^{-2}$ d$^{-1}$) simulated with eddy flux tower rates measured, $W$ denotes the Willmott coefficient of agreement.

[Figure]

**Figure S8.** Comparison of net ecosystem exchange rates (NEE, in gC m$^{-2}$ d$^{-1}$) simulated with eddy flux tower rates measured, $W$ denotes the Willmott coefficient of agreement.

[Figure]

**Figure S9.** Comparison of net ecosystem exchange rates (NEE, in gC m$^{-2}$ d$^{-1}$) simulated with eddy flux tower rates measured, $W$ denotes the Willmott coefficient of agreement.

[Figure]

**Figure S10.** Comparison of net ecosystem exchange rates (NEE, in gC m$^{-2}$ d$^{-1}$) simulated with eddy flux tower rates measured, $W$ denotes the Willmott coefficient of agreement.

[Figure]

**Figure S11.** Comparison of net ecosystem exchange rates (NEE, in gC m$^{-2}$ d$^{-1}$) simulated with eddy flux tower rates measured.

[Figure]

**Figure S12.** Comparison of evapotranspiration fluxes (in mm d$^{-1}$) with EDDY-flux measurements, $W$ denotes the Willmott coefficient of agreement.

[Figure]

**Figure S13.** Comparison of evapotranspiration fluxes (in mm d$^{-1}$) with EDDY-flux measurements, $W$ denotes the Willmott coefficient of agreement.

[Figure]

**Figure S14.** Comparison of evapotranspiration fluxes (in mm d$^{-1}$) with EDDY-flux measurements, $W$ denotes the Willmott coefficient of agreement.

[Figure]

**Figure S15.** Comparison of evapotranspiration fluxes (in mm d$^{-1}$) with EDDY-flux measurements, $W$ denotes the Willmott coefficient of agreement.

[Figure]

**Figure S16.** Comparison of evapotranspiration fluxes (in mm d$^{-1}$) with EDDY-flux measurements, $W$ denotes the Willmott coefficient of agreement.

[Figure]

**Figure S17.** Comparison of evapotranspiration fluxes (in mm d$^{-1}$) with EDDY-flux measurements, $W$ denotes the Willmott coefficient of agreement.

[Figure]

**Figure S18.** Comparison of evapotranspiration fluxes (in mm d$^{-1}$) with EDDY-flux measurements, $W$ denotes the Willmott coefficient of agreement.

[Figure]

**Figure S19.** Comparison of evapotranspiration fluxes (in mm d$^{-1}$) with EDDY-flux measurements, $W$ denotes the Willmott coefficient of agreement.

[Figure]

**Figure S20.** Comparison of evapotranspiration fluxes (in mm d$^{-1}$) with EDDY-flux measurements, $W$ denotes the Willmott coefficient of agreement.

[Figure]

**Figure S21.** Taylor diagram of the spatial patterns (national mean yields) of wheat productivity. The performance of *LPJmL5* is depicted as the red circlecross (⊕) and should be compared to the *LPJmL3.5* simulations, depicted as stars (✳) in yellow (un-calibrated) and blue (calibrated). This figure was produced by the online crop model evaluation tool of **?**.

[Figure]

**Figure S22.** Taylor diagram of the spatial patterns (national mean yields) of maize productivity. The performance of *LPJmL5* is depicted as the red circlecross (⊕) and should be compared to the *LPJmL3.5* simulations, depicted as stars (✳) in yellow (uncalibrated) and blue (calibrated). This figure was produced by the online crop model evaluation tool of **?**.

[Figure]

**Figure S23.** Taylor diagram of the spatial patterns (national mean yields) of rice productivity. The performance of *LPJmL5* is depicted as the red circlecross (⊕) and should be compared to the *LPJmL3.5* simulations, depicted as stars (✳) in yellow (un-calibrated) and blue (calibrated). This figure was produced by the online crop model evaluation tool of **?**.

[Figure]

**Figure S24.** Taylor diagram of the spatial patterns (national mean yields) of soybean productivity. The performance of *LPJmL5* is depicted as the red circlecross (⊕) and should be compared to the *LPJmL3.5* simulations, depicted as stars (✳) in yellow (un-calibrated) and blue (calibrated). This figure was produced by the online crop model evaluation tool of **?**.

[Figure]

**Figure S25.** Wheat yield simulations (in t fresh matter (FM) ha$^{-1}$) for the 10 top-producing countries for the carbon-only LPJmL3.5 version (*LPJmL35*), the version with N limitation (*LPJmL5*)and with unlimited N supply (*LPJmL5-nL*). The residuals plotted are the detrended observed and simulated yields.

[Figure]

**Figure S26.** Rice yield simulations (in t fresh matter (FM) ha$^{-1}$) for the 10 top-producing countries for the carbon-only LPJmL3.5 version (*LPJmL35*), the version with N limitation (*LPJmL5*)and with unlimited N supply (*LPJmL5-nL*). The residuals plotted are the detrended observed and simulated yields.

[Figure]

**Figure S27.** Soybean yield simulations (in t fresh matter (FM) ha$^{-1}$) for the 10 top-producing countries for the carbon-only LPJmL3.5 version (*LPJmL35*), the version with N limitation (*LPJmL5*) and with unlimited N supply (*LPJmL5-nL*). The residuals plotted are the detrended observed and simulated yields.